# The *Drosophila* RNA Helicase Belle (DDX3) Non-Autonomously Suppresses Germline Tumorigenesis Via Regulation of a Specific mRNA Set

**DOI:** 10.3390/cells9030550

**Published:** 2020-02-26

**Authors:** Alexei A. Kotov, Baira K. Godneeva, Oxana M. Olenkina, Vladimir E. Adashev, Mikhail V. Trostnikov, Ludmila V. Olenina

**Affiliations:** Institute of Molecular Genetics, Russian Academy of Sciences, 2 Kurchatov Sq., 123182 Moscow, Russia

**Keywords:** DDX3 RNA helicase, tumorigenesis, *Drosophila* testes, germline stem cells, cyst cells, translational regulation, CLIP-seq analysis

## Abstract

DDX3 subfamily DEAD-box RNA helicases are essential developmental regulators of RNA metabolism in eukaryotes. *belle*, the single *DDX3* ortholog in *Drosophila,* is required for fly viability, fertility, and germline stem cell maintenance. Belle is involved both in translational activation and repression of target mRNAs in different tissues; however, direct targets of Belle in the testes are essentially unknown. Here we showed that *belle* RNAi knockdown in testis cyst cells caused a disruption of adhesion between germ and cyst cells and generation of tumor-like clusters of stem-like germ cells. Ectopic expression of *β-integrin* in cyst cells rescued early stages of spermatogenesis in *belle* knockdown testes, indicating that integrin adhesion complexes are required for the interaction between somatic and germ cells in a cyst. To address Belle functions in spermatogenesis in detail we performed cross-linking immunoprecipitation and sequencing (CLIP-seq) analysis and identified multiple mRNAs that interacted with Belle in the testes. The set of Belle targets includes transcripts of proteins that are essential for preventing the tumor-like clusters of germ cells and for sustaining spermatogenesis. By our hypothesis, failures in the translation of a number of mRNA targets additively contribute to developmental defects observed in the testes with *belle* knockdowns both in cyst cells and in the germline.

## 1. Introduction

DEAD-box RNA helicases are important developmental regulators of gene expression. They operate by remodeling the local secondary structure of RNAs and RNA-protein complexes in an ATP-dependent manner to provide subsequent association of RNA-binding proteins to their RNA targets. This implicates a crucial role of DEAD-box RNA proteins in regulating ribonucleoprotein (RNP) complexes. In line with this, DEAD-box RNA helicases participate in practically every stage of RNA metabolism, including transcription, processing, translation, ribosome biogenesis, splicing, RNA localization, RNA decay, and turnover [1,2,3]. DEAD-box proteins of the large DDX3 subfamily are found to be conserved from yeast to mammals. The human genome contains two paralogous *DDX3* genes, one located in the X chromosome (*DDX3X)* and another in the non-recombining region of the Y-chromosome (*DDX3Y* or *DBY*) [4]. Expression of *DBY* is restricted to male germ cells [5]. DBY function is required for early stages of testis development in human fetal germ cells, including the transition from primordial germ cells to prospermatogonial cells [6]. Deletions encompassing the *DBY* gene lead to severe azoospermia and cause Sertoli cell-only syndrome (SCOS), which is characterized by complete germ cell loss in the testis seminiferous tubules along with preservation of somatic Sertoli cells [7,8,9,10].

DDX3X (henceforth DDX3) is found to be expressed ubiquitously in different human tissues. Currently the molecular function of DDX3 in human tumorigenesis is the subject of intensive research all over the world. Altered DDX3 expression is observed in tissue biopsies of patients with lung, breast, colon, liver, brain, and skin cancers, and in those with leukemia. Increased expression of DDX3 is often associated with aggressive phenotypes of human malignant tumors [11,12,13]. It was shown that DDX3 can function both as oncogene or tumor-suppressor depending on the type of cancer. This is associated with the predominant signaling pathways acting in the particular tissues, in which DDX3 is involved. Some of these pathways have been identified to date: the DDX3-E-cadherin pathway; the p53-DDX3-p21 pathway; the Wnt signaling cascade; the p53-MDM2-Slug-E-cadherin cascade, and others [13]. A more deeply studied case is DDX3 involvement in cancer as a component of the Wnt signaling cascade in the modulation of cell adhesion, motility, and metastasis. DDX3 positively regulates translation of Rac1 factor, which protects the key effector of the Wnt cascade, β-catenin, from ubiquitin-mediated proteasomal degradation [14,15]. DDX3 also acts as a regulator activity of CK1ε kinase which phosphorylates factor Dishevelled, a component of the Wnt-cascade, leading to the disruption of the β-catenin destruction complex and translocation of stabilized β-catenin to the nucleus, where it activates transcription of target genes [16]. Knockdown of DDX3 leads to a reduced level of Rac1 translation and destabilization of β-catenin that causes transcriptional repression of β-catenin target genes [14].

*belle (bel)* is the single ortholog of *DDX3* in *Drosophila melanogaster*. It has been shown that *bel* is essential for fly viability [17]. Belle protein is expressed ubiquitously and possesses extremely high functional pleiotropy in fly tissues, being involved in the ecdysone-triggered transcriptional cascade during metamorphosis [18]; in the small RNA-based silencing mechanisms [19,20]; in the Notch-induced differentiation of ovarian follicle cells and establishment of correct anterior–posterior polarity of the developing oocyte [21]; in the proper chromosome segregation during mitotic anaphase in S2 cells [22]; in the circadian rhythmicity in *Drosophila* neurons [23]; and in the cell cycle regulation in somatic tissues and the germline [24,25,26]. RNA helicase Belle is shown to regulate the translation of a set of specific mRNAs in an ATP-dependent manner [18,26,27,28]. As such, Belle is found as a member of a large ribonucleoprotein complex which includes Smaug, Cup, Me31B, Trailer hitch, PABPC, eIF4E proteins, and the CCR4-Not complex interacting with *nanos* mRNA in early embryos. This complex provides translational repression of the target mRNAs by shortening their poly(A)-tails [28]. Belle function is essential for male and female fertility [17,25,26]. Defects in cytokinesis during male meiosis and subsequent disorganization of spermatid bundles have been observed for hypomorph *bel* alleles [17]. Belle is required cell-autonomously for mitotic progression and survival of germline stem cells in the testes. Deficiency of *bel* leads to severe depletion of early germ cells via apoptosis in the testes of adult males, while somatic cyst cells and hub cells are still maintained [25]. This developmental disorder recapitulates the SCOS phenotype in humans with *DBY* gene deletions and also leads to severe male sterility [25,29]. Belle functions as an upstream regulator of expression of mitotic cyclins A and B both in the testes and ovaries [25,26]. However, regulatory and mechanistic actions of Belle in spermatogenesis are still largely understudied.

The maintenance of spermatogenesis in sexually reproducing organisms is a crucial condition for the species survival. The testes of *Drosophila melanogaster* have proved to be a valuable model system for studying the complex network regulating the processes of gametogenesis. The testis apical tip of a *Drosophila* male is occupied by a structure composed of somatic cells, called the hub, which serves as a niche for supporting two populations of stem cells, germline stem cells (GSCs) and somatic cyst stem cells (CySCs) (Figure 1A). A self-renewing division of a GSC produces a new GSC and a goniablast (spermatogonium). The division of a CySC leads to the emergence of a new CySC and a cyst cell. A pair of cyst cells surrounds the goniablast, generating a functional unit of spermatogenesis - the cyst. The tight interactions and signal exchange between somatic cyst cells and germ cells in the cyst are required for the differentiation of germ cells during spermatogenesis [30,31,32]. Spermatogonial cells in the cysts undergo four consecutive synchronous mitotic divisions, eventually producing a 16-cell cyst of interconnected primary spermatocytes surrounded by two non-dividing somatic cyst cells. Spermatocytes go through a prolonged growth phase and enter meiosis resulting in the generation of 64 haploid spermatids per cyst. Spermatids undergo subsequent development and individualization and are stored in seminal vesicles as mature sperm [33]. Somatic cyst cells are considered as functional analogues of human Sertoli cells and, like Sertoli cells, they are involved in the regulation of all stages of germ cell differentiation. The main stages of spermatogenesis are conservative in *Drosophila* and human [34]. A regulatory network ensuring a tight control of the balance between GSC renewal and differentiation is based mainly on translational activation and repression of multiple transcribed mRNAs to provide proper fate decisions and maintenance of spermatogenesis [34].

Here we dissected the role of RNA helicase Belle (DDX3) in somatic cyst cells of the testes and showed that Belle function in cyst cells is required non-autonomously for proper differentiation of early germ cells; *RNAi bel* knockdown *(belKD)* in cyst cells led to the segregation of early germ cells and cyst cells from each other and the accumulation of over-proliferating tumor-like clusters of early germ cells. We found that ectopic expression of *β-integrin* in cyst cells rescued the differentiation of early germ cells in the *belKD* testes, indicating the importance of integrin adhesion complexes in the tight interaction between somatic cyst cells and germ cells in the cyst. To further address the importance of Belle activity in testis development and spermatogenesis maintenance, we performed cross-linking immunoprecipitation and sequencing (CLIP-seq) analysis using anti-Belle antibodies. Our data revealed that Belle interacts with multiple mRNAs both in somatic and germ cells of the testes. By comparing our data with published results of screen studies, we selected a set of genes, knockdowns of which in the testes led to pathological phenotypes similar to *belKDs* in somatic cyst cells or in the germline; among these genes are *not1*, *caf1-55,* and also several genes encoding proteins involved in ubiquitin-dependent protein degradation. Thus, the set of candidate Belle targets includes mRNAs of factors that are essential for preventing tumorigenesis of early germ cells and ensuring proper development of mature gametes.

## 2. Materials and Methods

### 2.1. Fly Stocks and Genetics

Germinal tissues of adult *D. melanogaster* males generally raised at 23 °C were used (if not indicated differently). Knockdown of *bel* by RNAi was performed by using a *UAS*-driven *belle*-specific hairpin. To generate flies with *bel* knockdown in the germline, we crossed *UAS-bel RNAi* (VDRC #6299, *w^1118^; P{GD1324}v6299/TM3*) males with *nos-Gal4* driver line #25751 (*P{w[+mC] = UAS-Dcr-2.D}1, w^1118^; P{w[+mC] = GAL4-nos.NGT}40*, Bloomington) females. To generate flies with *bel* knockdown in somatic cyst cells we crossed *UAS-bel RNAi* (VDRC #6299) males and females of the *tj-GAL4* driver line #104055 (*y*w*; P{GawB}NP1624/CyO, P{UAS-lacZ.UW14}UW14*, Kyoto) or *c587-GAL4* driver line (*P{GawB}C587, w*)* provided by Dr. M. Buszczak. In the latter case, parent stocks and crosses were maintained at 18 °C. The fly stock carrying the construct *UAS-gfp (yw; P{UAS-GFP.S65T} Myo31DFT2/SM1, CyRoi)* was used for generation of *c587-GAL4 > UAS-bel RNAi, UAS-gfp* flies. The fly stock carrying the construct *UAS-arm* (Bloomington #4782; *P{UAS-arm.S10}C, y^1^ w^1118^*) was used for “rescue” experiments for *GAL4 > UAS-bel RNAi* flies along with *C587-GAL4* driver line. The *yw* flies bearing the transgenic construct *p{UASbetaPS}* on the third chromosome under the control of *UAS* promoter sequence (kindly provided by Dr. K. Broudi [35] were crossed with flies carrying *C587-GAL4* driver for expression of *β-integrin* in cyst cells and maintained at 18 °C. To generate flies with Belle target gene knockdowns in somatic cyst cells we crossed *UAS-not1 RNAi* (VDRC #45463), *UAS-caf1-55 RNAi* (VDRC #105838), *UAS-rpn7 RNAi* (VDRC #101467), *UAS-Fs(2)Ket RNAi* (VDRC #22348), and *UAS-rhoI RNAi* (VDRC #51953) males with females of *tj-GAL4; UAS-Dcr2/TM6* driver line. To generate flies with *RNAi* knockdowns in germ cells we crossed males of these lines with females of *nos-GAL4* driver line. The *Df(1)w67c23(2)y* (designated here as *yw*) line was used as the wild-type line for CLIP-seq library preparation. We used males of line *y [1] v [1]; P{y[+t7.7] v[+t1.8] = TRiP.HMS00017}attP2 (#33623,* Bloomington) for raising *c587* > *UAS*-*white RNAi* flies.

### 2.2. Immunofluorescence Staining and Confocal Microscopy

Testes of adult males were dissected in phosphate-buffered saline (PBS) at 4 °C, washed with PBT (1×PBS, 0.1% Tween 20), and fixed in 3.7% formaldehyde and PBT for 30 min at room temperature. All the following procedures were carried out as described previously [36]. Staining was detected by laser scanning confocal microscopy using a Carl Zeiss LSM 510 META machine (Carl Zeiss). All images were taken with a z-resolution of 1 µm. The obtained pictures were imported into Imaris^®^ 5.0.1 (Bitplane AG) for subsequent processing. Counting of spectrosomes and the number of specific testis cells on the confocal images was carried out using Imaris software. Data were analyzed using Statistica version 10 (StatSoft) and RStudio.

### 2.3. Antibodies

The following antibodies were used for immunofluorescence staining: a mix of murine monoclonal anti-Lamin Dm0 ADL67.10 and ADL84.12 antibodies (Developmental Studies Hybridoma Bank, University of Iowa (DSHB)), 1:500; rabbit polyclonal anti-Lamin antibodies [37], 1:500; rat monoclonal anti-Vasa antibody (DSHB), 1:100; murine monoclonal anti-α-spectrin 3A9 antibody (DSHB), 1:200; murine monoclonal anti-Fasciclin III 7G10 antibody (DSHB), 1:25; murine monoclonal anti-Eyes Absent (Eya) Eya 10H6 antibody (DSHB), 1:100; guinea pig polyclonal anti-Traffic jam (Tj) antibodies [38], 1:5000; rabbit polyclonal anti-Piwi 2464 antibodies [39], 1:200; rabbit polyclonal anti-Vasa antibodies (R. Lehmann), 1:5000; murine monoclonal anti-Armadillo antibody N2 7A1 (DSHB), 1:25; and rabbit monoclonal anti-phospho-Histone H3Ser10 (H3S10ph) MC463 antibodies (Millipore), 1:500.

Alexa Fluor-labeled secondary goat anti-rat IgG, goat anti-rabbit IgG, goat anti-mouse IgG and goat anti-guinea pig IgG (Invitrogen) were used as secondary reagents at a dilution of 1:500. DAPI (4′,6-diamidino-2-phenylindole) (Sigma) was used for chromatin staining. 

For Western blot analysis, the following antibodies were used: murine monoclonal anti-β-actin antibody ab8224 (Abcam), 1:4000; rabbit polyclonal anti-Piwi 2464 antibodies [39], 1:2000; goat polyclonal anti-STAT dN-17 antibodies (Santa Cruz Biotechnology), 1:100; goat polyclonal anti-Hedgehog dD-12 antibodies (Santa Cruz Biotechnology); 1:100; rat monoclonal anti-Cubitus interruptus (Ci) 2A1 antibody (DSHB), 1:100; murine monoclonal anti-Rac1 antibody 610,651 (BD Transduction Laboratories), 1:1000; murine monoclonal anti-Gbb antibody 3D-6 (DSHB), 1:50; rabbit monoclonal anti-Dpp antibody (Santa-Cruz Biotechnology), 1:100; rabbit monoclonal anti-pERK antibody (Cell Signaling), 1:2000; rabbit monoclonal anti-pSmad antibody ab52903 (Abcam), 1:500; murine monoclonal anti-Not1 2G5 antibody [40], 1:250; rabbit polyclonal anti-Caf1-55 antibodies ab1766 (Abcam), 1:2000; and murine monoclonal anti-Rho1 antibody p1D9 (DSHB), 1:50. Samples were resolved by SDS-PAGE and blotted onto the polyvinylidene fluoride (PVDF) membrane Immobilon-P (Sigma). Alkaline phosphatase-conjugated anti-mouse, anti-rabbit, anti-rat, and anti-goat antibodies (Sigma) were used as secondary reagents at a dilution of 1:20,000. Blots were developed using the Immun-Star AP detection system (Bio-Rad Laboratories). All experiments were performed at least in triplicate with independent preparations of testis lysates.

### 2.4. RNA Extraction, Reverse Transcription, and Real-Time Quantitative PCR

For identification of the transcriptional level of basic adhesion complex components RT-PCR analysis was performed with primers shown below. Briefly, total RNA was isolated from sets of 50–100 pairs of dissected testes, using TRIzol Reagent (Invitrogen) according to the manufacturer’s recommendations. cDNA was synthesized using random hexamers and SuperScript II reverse transcriptase (Invitrogen). cDNA samples were analyzed by real-time quantitative PCR using the incorporation of SYTO-13 (Invitrogen). Thermal cycling consisted of 5 min at 95 °C, followed by 45 cycles of denaturation (94 °C, 20 s), annealing (64 °C, 20 s), extension (72 °C, 20 s), and a final extension of 5 min at 72 °C. All experiments were performed with at least three independent RNA samples; each sample was analyzed in duplicate. Statistical analysis for the group comparison was performed by a paired Student’s *t*-test. 

The following primers were used for PCR or RT-PCR:

*rp49* fw 5′-ATGACCATCCGCCCAGCATAC-3′, rev 5′-GCTTAGCATATCGATCCGACTGG-3′; *arm* fw 5′-TCAAGGCCGTCATTGGACTC-3′, rev 5′-TAAGCAGTCGCACCAGATGG-3′; *e-cad* fw 5′-GAATCCATGTCGGAAAATGC-3′, rev 5′-GTCACTGGCGCTGATAGTCA-3′; *β-int* fw 5′-CGCGGTGCTACCAAAACAC-3′, rev 5′-GAATCTGCTCAACTGTTATCGGA-3′. We used *rp49* (*rpL32*) as a loading control.

### 2.5. Fertility Tests

Individual males were placed with three virgin 5-to 7-day-old *yw* females for five days; after that, the parent flies were removed. Each tested group contained from 10 to 20 individual males. We analyzed the fertility of males of the four following genotypes: (1) *c587-Gal4;* (2) *c587-Gal4>UAS-βPs;* (3) *c587-Gal4>UAS-bel RNAi*; and (4) *c587-Gal4>UAS-βPs, UAS-bel RNAi.* Data for the *c587-Gal4* driver carrying line were used as wild-type control. Eclosed adult offspring were evaluated from days 10–18. Two independent experiments were performed.

### 2.6. Electron Microscopy

Testes of 0- to 3-day-old males were dissected and fixed in sodium cacodylate buffer with 2.5% glutaraldehyde at room temperature for 2 h. Samples were washed in sodium cacodylate buffer, fixed in 2% OsO_4_ for 2 h at 4 °C, washed, and, dehydrated by transferring through solutions of increasing alcohol concentration (30%, 50%, 70%, 90%, and 100%) for 15 min at room temperature followed by acetone for 15 min. Resin (Epon 812 with DDSA and MNA) infiltration was carried out according to the following protocol: resin-acetone (1:3), resin-acetone (1:1), resin-acetone (3:1) at room temperature for 1 h, resin alone for 12 h, and resin with catalysator DMP-30 at 37^◦^C for 24 h and at 60 °C for 48 h. Hardened blocks were cut on the Leica UC7 microtome (Leica) into 70 nm slices, contrasted by uranyl acetate (30 min) and lead citrate (5 min), and then analyzed using a transmission electron microscope (JEM-1011, Jeol, Akishima).

### 2.7. CLIP-seq Analysis

Freshly dissected testes of 0–3 day-old *yw* males (500 pairs per one library) were cross-linked on ice with UV radiation (254 nm) three times at 600 mJ/cm^2^ in the presence of 200 mkl cold PBS (in a 1 cm^2^ glass dish). The testes were transferred to 1.7-mL polypropylene microcentrifuge tubes and pelleted at 300 g at 4°C. The supernatant solution over the organs was carefully removed by a tip and cross-linked material was stored at -70°C until further use. Cells lysates were obtained using pre-chilled Dounce homogenizer on ice in a cold lysis buffer (containing 50 mM Tris-HCl, pH 8.0, 200 mM NaCl, 2 mM MgCl_2_, 1 mM DTT, 0.5% Nonidet P-40 (NP-40), in the presence of protease and phosphatase inhibitors (Sigma) in diethyl pyrocarbonate (DEPC)-treated water with the addition of Ribolock (Thermo Fisher Scientific)) by two series of 15 strokes of B pestle with a 1–2 min incubation between the series. The lysates were transferred to 1.7-mL polypropylene microcentrifuge tubes and were treated by RQ1 RNase-Free DNase (Promega) 40 U/mL for 10 min at 37°C with vortexing. To generate RNA fragments of appropriate length we treated our lysates by RNAse A for 7 min at 37°C. Immediately after that the lysates were supplemented by Ribolock. The lysates were subjected to centrifugation at 13,200 rpm at 4°C for 20 min. The supernatants were transferred to new tubes and diluted to protein concentration of 8–10 mg/mL with lysis buffer (containing 50 mM Tris-HCl, pH 8.0, 200 mM NaCl, 2 mM MgCl_2_, 1 mM DTT, 0.5% Nonidet P-40, protease and phosphatase inhibitors (Sigma), Ribolock). We used 150 µL of mixed 50%-slurry of anti-mouse IgG beads (Invitrogen) for each tube of the cross-linked lysate. The beads were previously washed two times with 500 µL cold phosphate-buffered saline (PBS) supplied by Tween 20 (1 × PBS, 0.1% Tween 20 (PBST)) and incubated with 8 µL anti-Belle antibodies (described in [25]) for CLIP library preparations or 8 µL normal mouse serum for negative control in 200 mkl PBST during 20 min at room temperature on a rotator. The antibody-coated beads were washed and pre-equilibrated with lysis buffer. Then the clarified lysates were added to the experimental and control tubes. After incubation for 40 min at RT on a rotator, washing was performed four times with 1 mL washing buffer NT2 (50 mM Tris-HCl, pH 8.0; 300 mM NaCl; 1 mM MgCl_2_; 1 mM DTT; 0.05% NP-40) and once with 200 µL (without Ribolock). The beads were pre-equilibrated with buffer for T4 RNA ligase 2 truncated (New England Biolabs (NEB)). Dephosphorylation of RNA 3′-ends was performed *on-bead* by Calf Intestinal Alkaline Phosphatase (Roche) 1U/µL for 10 min at 37°C with frequent agitation. After that the beads were washed by 1 mL NT2 buffer with Ribolock two times. A previously pre-adenylated 3′-adapter (5′DNA Adenylation Kit (NEB) was used for preparation) was ligated with RNAs *on-bead* by T4 RNA ligase 2 (NEB) 400 U in 30 µL of ligation reaction 2 h at 25°C on a thermal mixer. The beads were washed three times by 1 mL NT2 buffer with Ribolock. 5′-end RNA labeling was done by treating the samples with 4 µL PNK enzyme (10 U/mkl; NEB), 1× PNK B buffer (NEB) and 2 µL γ32P-ATP (10 mCi/mL) for 10 min at 37°C with mixing at 1000 rpm for 15 s after each 3 min period of incubation. After that 10 µL of cold 1 mM ATP was added to the beads and incubation was continued for 5 min. The beads were washed three times with NT2 buffer. Belle-RNA complexes were eluted with 30 µL 2x sample buffer supplemented by 200 mM DTT 5 min at 70°C, and the complexes were resolved in 8% Bis-Tris gels using 3-(N-morpholino)propanesulfonic acid (MOPS)-based running buffer (50 mM MOPS, 50 mM Tris base, 1 mM EDTA, 0.1% SDS, DEPC-treated water). The complexes were transferred from the gel into a nitrocellulose membrane (BA-85 S&S). The membrane was exposed to a storage phosphor screen for 1–2 h. Storage phosphor screen was scanned using Typhoon FLA700 Phosphoimager (GE Healthcare), and an image was obtained. Fragments of the membrane containing the main radioactive signals over 85 kDa were cut, fragmented with a clean scalpel blade, transferred to tubes, and incubated with 100 µL proteinase K solution (4 mg/mL proteinase K dissolved in buffer PK (100 mM Tris-HCl, pH 7.5, 10 mM EDTA and 50 mM NaCl) for 20 min at 37 °C with agitation at 1000 rpm. After that 100 µL PK/7M-urea buffer were added to each tube and incubated for additional 20 min at 37 °C at 1000 rpm. Then 200 µLTrisol LS (Invitrogene) for RNA isolation and 200 µL chloroform were added to each tube and mixed vigorously for 30 s. After incubation 5 min at 4 °C the tubes were centrifuged at 13,000 g and 4 °C for 10 min. The upper phases were transferred to new microcentrifuge tubes and precipitated in the presence of 300 mM sodium acetate, 1 µl Glycoblue (from stock 15 mg/mL, Thermo Fisher Scientific), and 2.5× volume of EtOH overnight at −20 °C. After that ligation of 5′-adapter RA5 was performed using 1 µL T4 RNA ligase (NEB) in manufacturer’s buffer supplemented by 1mM ATP solution for 2 h at 37 °C with subsequent precipitation. cDNA libraries were prepared after reverse transcription (SuperScript III Reverse Transcriptase kit (Thermo Scientific Fisher)) and subsequent PCR tests for optimal amplification with forward RP1 primer and reverse Illumina index primer and HS-Taq DNA pol (Thermo Fisher Scientific). The CLIP libraries were sequenced on the Illumina HiSeq2500 platform. CLIP sequencing data has been deposited to the NCBI GEO under accession number GSE137956.

### 2.8. Computational Analysis of CLIP Libraries

Preprocessing of the CLIP libraries involving adapter removal, filtering of reads according to their quality (MAPQ score > 15) and collapsing of reads with exact matching sequences was performed using bioinformatics tools that based on the Galaxy framework [41]. Reads that were duplicated due to PCR amplification step were collapsed in silico to one read by removing excess of identical sequences associated with the same barcode. Eventually each read represents an individual RNA molecule cross-linked with Belle. Then the reads having survived the preprocessing stage were mapped on the last assembly of *Drosophila melanogaster* genome *dm6* (UCSC Genome Browser, https://genome.ucsc.edu/) using Bowtie 2 [42] or TopHat [43] algorithms. The reads were mapped allowing unique hits and up to two mismatches. The reads were annotated as falling in the following categories: 5′-UTR; 3′-UTR; CDS; introns; ncRNAs including snoRNAs, snRNAs, small RNAs and transposable elements; unannotated regions of genome. Peak calling to determine specific signals over background reads were performed using Piranha software [44] by the zero-truncated negative binomial likelihoods method. The whole *Drosophila* genome was split into 30-nt intervals (bins) and coverage in these windows was counted for maximum coverage greater than or equal to the 0.87 cut-off (selection conditions *p* > 0.87, bin is equal 30 nt). Only peaks with the coinciding or overlapping genome coordinates determined by Piranha in both CLIP libraries were selected as high confidence ones (Appendix A). The peaks that were determined by Piranha in the above-mentioned conditions for only one CLIP library and present in the second CLIP library as at least two overlapping reads were considered as medium confidence ones (Appendix A). Pearson’s correlation analysis for comparison of CLIP1 and CLIP2 libraries was performed using Libre Office software. To analyze the distribution of the reproduced peaks in protein-coding mRNAs, four hierarchical categories were created, as follows: 5’UTRs > 3’UTRs > CDS > introns. Peaks were manually reviewed using *dm6* genome assembly in a local mirror of the UCSC genome browser. To predict and expose the functional networks of proteins encoded by mRNAs found in the CLIP-seq libraries as candidate Belle targets we used the DAVID v6.8 [45] and STRING [46] databases.

## 3. Results

### 3.1. Functioning of Belle in Cyst Cells is Required for Differentiation of Early Germ Cells

Recently we have shown that RNA helicase Belle (DDX3) is a critical cell-autonomous factor for survival and divisions of GSCs and spermatogonial cells in the *Drosophila* testes [25]. Complete loss of germ cells occurs in the testes of newly eclosed *bel^6/neo30^* mutants, as well as in the germline-specific *RNAi*-mediated knockdown of *bel* (*RNAi belKD)*. In contrast, in *RNAi belKD* in somatic testis cells (the cyst cells) using the *tj-GAL4* driver leads to the maintenance of the germline content. However, we observe that about 53% analyzed *tj-GAL4>UAS-bel RNAi* testes exhibit the accumulation of multiple small germ cells arranged in tight clusters [25].

Here we found that the usage of another soma-specific driver, *c587-GAL4,* for *RNAi belKD* also had a profound effect on testis morphology and gametogenesis. This driver is mostly active in CySCs and early cyst cells of the testes [47]. Consistent with our previous observations 55.9% (57 of 102) of the analyzed testes of newly eclosed *c587-GAL4>UAS-bel RNAi* males were significantly reduced in size and filled with tightly packed clusters of small Vasa-positive germ cells (Figure 1B,D). These testes practically did not contain spermatocytes and germ cells of subsequent spermatogenesis stages. Another fraction of the *RNAi belKD* testes (38.2%, 39 of 102 cases) exhibited the so-called mosaic phenotype with several small germ cell clusters in the neighborhood with spermatogonial cells and spermatocytes (Figure 1B,E). We observed only rare cases of a wild-type-like phenotype for the *RNAi belKD *testes (5.9%, 6 of 102 cases), whereas all analyzed control testes displayed a wild-type phenotype (100%, *n* = 80) (Figure 1B,C).

To examine the relationship between germline and somatic cyst cells in *RNAi belKD* testes, we used antibodies against Traffic jam (Tj), the soma-specific transcription factor that marks nuclei of CySCs and early cyst cells [38]. In the control *c587-GAL4 *testes we observed a wild-type distribution of Tj-stained CySCs (in the next row from the hub behind GSCs) and cyst cells which enveloped mitotically active spermatogonia in the testis anterior (Figure 1C). However, *RNAi belKD* testes contained a certain excess of Tj-stained early cyst cells which were not associated with germ cells (Figure 1D,E). The number of Tj-positive cyst cells in the *c587-GAL4>UAS-bel RNAi *testes was found to be insignificantly greater in comparison with the ones in the control testes (average number is 76.5 ± 48.3 cells, *n* = 68 versus 56.4 ± 13.4 cells, *n* = 51 in control; *p* = 0.073 (Wilcoxon/Mann–Whitney test)) (Figure 1F). We also analyzed the testes of *c587-GAL4>UAS-white RNAi* male as independent control (Appendix A); however, the number of Tj-positive cyst cells also did not differ significantly.

Using flies with the *c587-GAL4*-induced transgenic *UAS-gfp* construct along with *UAS-bel RNAi *we clearly established that GFP-marked somatic cyst cells mainly did not encapsulate early germ cells in the testes, but rather located separately from germ cell clusters being pushed to the testis shell (Appendix A). These somatic cells had a round shape in contrast to their flat elongated shape in the control testes *c587-GAL4>UAS-gfp* (Appendix A). The nuclei of Tj-positive cyst cells in the *belKD* testes also conform to a spherical shape; these nuclei have an increased volume and Tj protein exclusion from a certain internal space, presumably from the nucleolus (Figure 1D,E). The significance of changes in nuclear morphology is not clear to date and requires further researches. We proposed that *RNAi belKD* in cyst cells led to a cessation of their internal differentiation program.

The marker of mature cyst cells Eyes Absent (Eya) [30] is expressed at a high level in cyst cells of spermatocyte-containing cysts of the control testes (Appendix A). We found a severe decline in the number of Eya-positive mature cyst cells in the *c587-GAL4>UAS-bel RNAi* testes compared with control testes of *c587-GAL4* males (3.36 ± 7.95 cells, *n* = 25 versus 46.3 ± 18.21 cells, *n* = 10, *p* = 4.528∙× 10^−6^ (Wilcoxon/Mann–Whitney test) (Appendix A; Figure 1G).

These data indicate that Belle deficiency in somatic cyst cells does not lead to severe disturbances in the maintenance and proliferation of CySCs. Nevertheless, *belKD* in early cyst cells often leads to a cell-autonomous effect of their accumulation. Most of Tj-stained cyst cells failed to envelope germ cells and instead are located separately. They do not undergo differentiation to mature cyst cells. In summary, our data indicate that *RNAi belKD *in cyst cells generally lead to the segregation of early germ cells and cyst cells from each other, the accumulation of tightly packed clusters of small germ cells near the testis apical tip, and certain enrichment in early cyst cells for subset of the *RNAi belKD* testes. 

### 3.2. Proliferation Activity of Germ Cells in the Clusters

To study the peculiarities of proliferation of germ cells in the clusters of the *RNAi belKD* testes, we analyzed the distribution of spectrosomes and fusomes in the testes using antibodies against α-spectrin. Spectrosomes are spherical organelles that are inherent for GSCs and goniablasts, whereas fusomes are specific branched structures penetrating all spermatogonia or spermatocytes within an individual cyst through syncytial ring canals [33] (Figure 1A). Control testes of *c587-GAL4* males displayed a wild-type pattern of spectrosomes and fusomes (Figure 2A). In contrast, in the *RNAi belKD* testes, germ cells in the clusters contained mainly dot-like spectrosomes which are characteristic structures of GSCs (Figure 2B). We also found that germ cells in the clusters maintained proliferation activity. Immunostaining of testis preparations with mitotic marker phospho-Ser10H3 (PH3) antibodies revealed that germ cells in the clusters divided independently of each other and asynchronously (Figure 2D) in contrast to the synchronous divisions of spermatogonial cells in the cysts of the control testes (Figure 2C). The number of individual asynchronous divisions of germ cells in the *RNAi belKD* testes was significantly higher, and reached 1.65 divisions per testis on the average compared with the control ones, where their number was only 0.27 per testis (*p* = 0.0093, Wilcoxon/Mann–Whitney test) (Figure 2E). At the same time, the frequency of synchronous signals in germ cells did not significantly differ in the experimental (0.45 signals per testis, *n* = 20) and control testes (0.44 signals, *n* = 18) (*p* = 0.8886) (Figure 2E). Because the meiotic cell division in the wild-type testes appears to be a rapid process, at any given moment we can see only a small number of dividing cells. Note that divisions of individual germ cells in the clusters occurred independently of their distance from the hub (Figure 2D).

The founder protein of the PIWI subfamily of the ARGONAUTE family Piwi is known to participate in co-transcriptional repression of transposons and contribute to the control of GSC proliferation in the *Drosophila* testes [48,49]. Analyzing expression of Piwi we observed that in the wild-type testes nuclear protein Piwi brightly stains hub cells at the topmost part of the testes. Strong Piwi staining is also detected in GSCs that tightly adhere to the hub, as well as in their immediate daughters, goniablasts. Expression of Piwi sharply drops in germ cells at later stages of differentiation. Piwi is also detected in CySCs and cyst cells throughout all premeiotic stages (Appendix A) according to previous papers [36,48,49]. However, we found a significant overexpression of Piwi in the *RNAi belKD* testes in comparison with the control testes by immunostaining and Western blot analysis (Appendix A). In the *RNAi belKD* testes, Piwi strongly marked nuclei of all germ cells of the clusters (Appendix A) and was also expressed in somatic cyst cells (Appendix A, yellow arrowheads). The pattern of Piwi expression additionally confirms that germ cells in the clusters are of GSC-like nature.

Taken together, our results indicate that germ cells in the clusters retained stem-like properties and supported asynchronous mitotic divisions with complete cytokinesis, independently of proximity to niche structure, the hub. Germ cells in the clusters expressed GSC markers and did not undergo subsequent differentiation.

### 3.3. Ectopic Overexpression of A Transgenic Arm^s10^ COPY in Cyst Cells Did Not Restore the Belle Knockdown Phenotype in the Testes

We speculated that the observed phenotypic abnormalities in the cyst cell *RNAi belKD* testes, such as the generation of tumor-like clusters of non-differentiated germ cells, could be caused by a disruption of intercellular adhesion between early germ and somatic cyst cells. Cell adhesion is mediated by specialized adhesion molecular complexes that are exposed on the surface of the interacting cells. However, our RT-qPCR analysis did not detect significant differences in the expression of transcripts encoding proteins of basic testis adhesion complexes, *E-cadherin* (*E-cad*), *arm* (*armadillo*, *β-catenin* in *Drosophila*), and *β-integrin,* in the testes with *RNAi belKD* in cyst cells compared to the control (Appendix A). Western blot analysis also showed that total levels of E-cad, Arm, and β-integrin proteins in the *RNAi belKD* testes did not change significantly from those in the control testis (*p* > 0.05, Student’s test) (Appendix A). We assumed that failures in the cyst formation observed in the *RNAi belKD* testes may be associated with improper traffic or exposure of the adhesion complexes on the surface of early cyst cells.

To examine whether the deficiency of *Belle* function in somatic cyst cells actually leads to the disruption of the formation of cadherin-catenin adhesion complexes we used the fly line carrying a transgenic construct *UAS-arm^S10^* [50] encoding a non-degradable form of Arm. We generated flies with the *c587-GAL4*-induced *UAS-arm^S10^* transgenic construct along with *UAS-bel RNAi* and confirmed an increased level of Arm protein in their testes (Appendix A). However, the testes of experimental males did not differ morphologically from the testes with *RNAi belKD* alone and contained the clusters of small germ cells having escaped their enveloping by somatic cyst cells (Appendix A). Thus, ectopic overexpression of a transgenic *arm^S10^* copy in cyst cells was not able to restore wild-type testis phenotype.

### 3.4. Ectopic Expression of β-Integrin Rescued Early Stages of Spermatogenesis in RNAi belKD Testes

Integrin adhesion complexes have been shown to play a pivotal role in the balance and competition between CySCs and GSCs for niche occupancy [51]. We examined the influence of integrin-based adhesion complexes on the repression of the tumor-like germline clusters in the background of *RNAi belKD* in cyst cells (Figure 3A–D, Appendix A). We found that ectopic overexpression of *β-integrin* (encoded by *myo* (*βPS*) gene) led to a significant rescue of the germline tumor phenotype. In total, 61.2% (*n* = 30) experimental testes exhibited wide-type size and morphology including the proper formation and maintenance of the cysts and the differentiation of premeiotic germ cells (Figure 3D). Another fraction of the testes, 38.8% (*n* = 19), exhibited a mosaic phenotype characterized by the presence of small germ cell clusters along with the cysts of spermatogonial cells and spermatocytes (not shown). We found that the level of Piwi, marker of GSCs, returned to wild-type and the marker of spermatocytes βNACtes [52] was restored to the wild-type level indicating the maintenance of the spermatocyte population in the “rescued” testes (Figure 3E). In the wild-type testes the expected number of spectrosomes does not exceed 30. We found that in most of the experimental testes (*belKD + β-int*) the number of spectrosomes did not exceed 30 per testis (Figure 3F). In sum, these data confirm the restoration of premeiotic stages of spermatogenesis. Thus, expression of integrin adhesion complexes is important for the maintenance of interaction between cyst cells and germ cells in the testes and for the prevention of germ cell tumor formation. However, the rescue of early stages of spermatogenesis owing to the expression of an additional copy of *β-integrin* did not lead to the restoration of male fertility (Appendix A). Using electron microscopy, we observed developmental defects in the experimental testes (*belKD + β-int*) at the stage of spermatid elongation (Appendix A). Apparently, the ectopic expression of *β-integrin* in early cyst cells does not allow rescue of the post-meiotic stage of spermatogenesis in the testes with *belKD* in cyst cells.

### 3.5. CLIP-seq Analysis for Identification of Belle mRNA Targets 

To identify direct mRNA targets of Belle in the testes, we performed cross-linking immunoprecipitation assay with subsequent deep sequencing (CLIP-seq) as described in Materials and Methods. We used UV-cross-linked testes of young *yw* males and anti-Belle antibodies raised previously [25] for the preparation of two biological replicates of CLIP-libraries. The recovery of cross-linked target RNAs in RNA-Belle complexes was monitored by autoradiography (Figure 4A). The recovery of endogenous Belle protein in the immunoprecipitation procedure was monitored by Western blot analysis (Appendix A). Protein-RNA complexes were practically absent when anti-Belle antibodies were replaced with normal mouse IgG, indicating that the Belle-RNA complexes immunoprecipitated in these experiments were specific complexes (Figure 4A; Appendix A). The resulting cDNA libraries were barcoded and subjected to high-throughput sequencing. For the two CLIP-seq experiments we obtained 8.1 × 10^6^ and 4.8 × 10^6^ raw reads. However, we failed to amplify background material in the control experiment with normal mouse IgG instead of anti-Belle antibodies (not shown). This indicates a high signal-to-noise ratio for the Belle CLIP-libraries. Bioinformatics data analysis was performed as described in Materials and Methods. The number of reads for the Belle CLIP-libraries before and after processing and genome mapping are presented in (Figure 4C). Then, peak calling was performed using Piranha software [44] to identify a list of Belle target RNAs (Appendix A). Two biological replicates of the CLIP-libraries showed high reproducibility in the numbers of unique-mapped reads per gene in the CLIP experiments (Pearson’s correlation coefficient 0.88, Figure 4B). Only the peaks with coinciding or overlapping genome coordinates found in both of the CLIP libraries were selected for subsequent analysis. The distribution of unique peaks revealed that 60% of them maps to annotated regions of the genome (Figure 4D). Across transcripts of protein-coding genes, Belle was found to be enriched in general in coding sequences and both in 5′- and 3′-UTRs (see Appendix A for several examples of peaks mapped in 5′- and 3′-UTRs of protein-coding genes), and exhibited comparatively low binding to introns, 28.6% (Figure 4E). This circumstance points out that Belle interacts preferably with mature mRNAs in the testes. However, according to published data, DDX3 helicases can be involved in a wide range of intracellular processes associated with RNA metabolism, including transcription and splicing [3]. Thus, we cannot exclude the participation of Belle in mRNA processing in the nucleus. Overall, we mapped Belle binding sites in mature transcripts of more than 300 protein-coding genes (Appendix A).

Gene ontology (GO) analysis using DAVID software to search for overrepresented GO categories in comparison to the entire fly protein-coding transcriptome revealed that Belle targets were significantly enriched in genes encoding proteins which were involved in cell cytoskeleton organization, ubiquitin-dependent protein catabolic processes, sperm individualization, mitochondrial functions, intracellular signal transduction, transmembrane transport, and nucleic acid binding (Appendix A). GO functional annotation clustering revealed at least seven functional clusters among candidate Belle mRNA targets (Appendix A).

### 3.6. Belle is Involved in Translational Regulation of Multiple mRNA Targets

We speculated that Belle binding within 5′-UTRs may facilitate translation initiation by unwinding secondary structures of mRNAs owing to its helicase activity, whereas Belle interaction with 3′-UTR may contribute to either activation or repression of translation, according to previously published data [28]. The peaks determined in the coding regions suggest that Belle may participate in the elongation steps of translation as well [26,28].

Recently, several groups reported *RNAi*-mediated knockdown screens to identify candidate genes that are required in somatic cyst cells or in the germline of *D. melanogaster* testes for specific stages of spermatogenesis [53,54,55]. Having conducted comparative analysis of these datasets, we recovered a subset of 22 genes that coincide with the ones obtained in our CLIP-seq analysis as Belle targets (Appendix A). It has been shown that the *RNAi* knockdowns of 17 genes from this subset determined in our CLIP-seq analysis as potential targets of Belle regulation lead to similar phenotypes as *belKD* in cyst cells or in the germline, according to the results of at least one screen (Figure 5A, Appendix A). Among them we selected four genes, *not1*, *rpn7*, *rho1,* and *caf1-55*, knockdowns of which result in the same phenotypic defects in the testes based on data from all three independent screens (Figure 5A).

We performed Western blot analysis of testis lysates for examination of expression levels of three proteins encoded by Belle-targeted transcripts in the testes of *c587-GAL4>UAS-bel RNAi* males, *nos-GAL4>UAS-bel RNAi* males, and the corresponding driver lines as controls. For Not1, the key component CCR4-Not deadenylation complex, we repeatedly observed a decrease in the intensity of bands corresponding to the major long (280-kDa) isoforms and the minor short (260-kDa) isoform of Not1 in the backgrounds of *RNAi belKDs* both in the soma and germline of testes (Figure 5B). We also found a reproducible reduction of nucleosome remodeling factor Caf1-55. However, we did not detect significant changes in the level of small GTPase Rho1.

To confirm these results and validate the relationships between *not1* transcripts and Belle in spermatogenesis, we carried out *RNAi* knockdown of *not1* using soma-specific driver *tj-GAL4*. Immunostaining of fixed testis preparations revealed that the control testes (*n* = 13) contained a wild-type array of cells at all stages of spermatogenesis, whereas the *not1KD* testes were reduced in size and exhibited obvious phenotypic disorders (Figure 5C–E). In most of the cases (*n* = 9), the germline content in the experimental testes was represented by only small undifferentiated cells assembled in tumor-like clusters with somatic Tj-positive cells located separately (Figure 5D). In the remaining testes (*n* = 5), clusters of small germ cells were observed along with differentiating germ cells, including spermatocytes (Figure 5E). Thus, *not1KD* in cyst cells reproduced the GSC tumor-like phenotype observed in the testes of males with *belKD* in cyst cells (see Figure 1D–E, Figure 2B and D). Taken together with the CLIP-seq data and Western blot analysis, these results indicate that Belle positively regulates translation of *not1* transcripts. We also confirmed that *RNAi* knockdowns of *caf1-55KD, rpn7* and *Fs(2)Ket* in cyst cells led to tumor-like germ cell cluster formation (Appendix A), but with relatively low penetrance compared to *not1KD.* Note that we performed pilot experiments using the *c587-Gal4* driver to reveal spermatogenesis defects in case of knockdowns of the selected genes. However, we did not observe any differences from the wild-type testis phenotype in these experiments (not shown), possibly due to low knockdown efficiency. For these reasons we used here the *tj-GAL4* driver line supplemented by an additional *dicer* copy.

Using the *nos-GAL4* driver we also performed *RNAi* knockdowns of *not1, caf1-55, rpn7, Fs(2)Ket* and *rhoI* in the germline (Appendix A). With aid of immunostaining we found a total loss of testis germ cells with the maintenance of somatic cells in cases of *RNAi* knockdowns of *caf1-55, rpn7,* and *Fs(2)Ket,* whereas the *RNAi* knockdown of *not1* led to a rapid decrease in both somatic and germ cells (Appendix A).

Using the STRING database [46], we clustered the genes from the full list of 22 candidate Belle CLIP-targets shared with the results of the screen studies [53,54,55] (Appendix A) for potential interactions and found a significant enrichment (*p*-value = 0.00449) of clusters of interacting genes with high confidence settings (Figure 5F; Appendix A). Among them the clusters of genes involved in protein ubiquitination and proteolysis, Golgi-Endoplasmic Reticulum vesicle transport of macromolecules, small GTPase Rho1 activity, and RNA processing factors were identified (Appendix A). 

To analyze whether any signaling pathways were modulated by *RNAi belKD* in somatic cyst cells of testes we performed a mini-screen using antibodies to key components of different pathways (Appendix A). We found that *belKD* in somatic cyst cells led to a significant decrease in the testes of transcription factor STAT92E, one of the main participants of the Janus kinase-signal transducer and activator of transcription (JAK-STAT) pathway, which is responsible for the attachment of GSCs and CySCs to the hub [31,56,57,58]. We often observed that germ cells in the clusters did not maintain tight contacts with the hub. Thus, the survival of germ cells in the clusters does not depend on the level of activation of the JAK-STAT pathway in them. We found that the level of the key effector of the Hedgehog pathway in cyst cells, the 155-kDa form of Ci [59], significantly decreased in the *belKD* testes (Appendix A). However, epidermal growth factor receptor (EGFR) pathway signaling activities in CySCs of the *belKD* testes were indistinguishable from those in the control testes, as revealed by the intensity of signals of the phosphorylated form of MAP kinase ERK, pERK, and of small GTPase Rac1, despite the fact that disruption of the EGFR pathway has been shown to exhibit a testis phenotype [60,61,62] similar to *belKD* in cyst cells. We determined that the level of the Glass bottom boat (Gbb) 55-kDa precursor, the signaling molecule of the main GSC self-renewal regulator bone morphogenetic protein (BMP) (transforming growth factor β (TGF-β)) pathway [63,64], was significantly increased in the testes of males with *belKD* in cyst cells (Appendix A). The level of the second BMP signaling molecule, Decapentaplegic (Dpp), was found to be unchanged. Note that only transcripts of *gbb*, but not *dpp*, were identified as direct targets of Belle in our CLIP-seq analysis (Appendix A). However, we did not detect significant changes in the level of phosphorylated Smad molecule (pSmad), a key component of the BMP signaling cascade in early germ cells in the *belKD* testes (Appendix A).

## 4. Discussion

The RNA helicase Belle is known to be essential for fly viability, fertility, and germline stem cell maintenance [17,25,26]. However, the regulatory and mechanistic roles of Belle in spermatogenesis are still largely obscure. Here we revealed function of RNA helicase Belle in the testes as a germ cell tumor suppressor acting *in-trans* in early cyst cells. We found that *RNAi*-induced *belKD* in cyst cells resulted in the generation of large tumor-like clusters of GSC-like germ cells. Germ cells in the clusters were maintained over time; they spontaneously and asynchronously divided, but did not undergo subsequent differentiation (Figure 1B–E, Figure 2B and D, Appendix A). Cyst cells failed to envelope these germ cells and instead were accumulated separately (Figure 2D, Appendix A). They were also arrested in their differentiation, since the transition of stage-specific differentiation markers did not occur (Figure 1F–G, Appendix A). Previously it has been shown that mutations in EGFR pathway components resulted in a similar tumor-like germ cell phenotype [60,61,62]. However, the molecular mechanisms underlying these spermatogenesis disorders are not clearly understood; and we found no evidence of a direct contribution of Belle to the regulation of the EGFR pathway in the testes (Appendix A).

During wild-type spermatogenesis, coordinated morphogenetic processes occur both in the germline and somatic cyst cells and their physical interactions are essential for maintenance and self-renewal of GSCs in the niche and also for differentiation of their progeny [30,31,32,34]. We determined that *belKD* in somatic cyst cells led to the disruption of contacts between early germ cells and cyst cells. According to our results, premeiotic stages of spermatogenesis in the *belKD* testes were significantly rescued by the ectopic expression of *β-integrin* (*βPS*), but not non-degradable *arm^S10^* (*β-catenin*) in early cyst cells (Figure 3 and Appendix A). It has been shown previously that integrin adhesion complexes in the testes are important for anchoring the hub on the surface of extracellular matrix [65,66] and also for the attachment of cytoplasmic protrusions of CySCs to the hub to obtain instructive signals that promote their maintenance and self-renewal divisions [51,67,68]. It is reported that SOCS36E protein represses the JAK-STAT signaling in CySCs cells. SOCS36E prevents these cells from displacing GSCs from their position near the hub by up-regulating integrin-based adhesion complexes [67,68]. The EGFR signaling pathway also contributes to the control of integrin expression in CySCs cells; and integrins play a crucial role in the regulation of competition between GSCs and CySCs for the niche occupancy [51]. Integrin heterodimeric complexes are known to mediate two-sided signaling. Intracellular signals cause conformational changes of integrin complexes, which contribute to their activation and increased affinity for extracellular ligands [69,70]. In turn, binding of integrin complexes with their extracellular ligands or clustering of integrin complexes on the cellular membrane contributes to conformational changes in their cytoplasmic domains, modulating their interactions with cytoskeletal proteins and activation of some signaling pathways [69,70,71]. We revealed here that transgenic overexpression of *β-integrin* provided significant restoration of early stages of spermatogenesis in the background of *RNAi belKD* in cyst cells (Figure 3). Our results confirm a more universal role of integrin complexes in CySCs in the maintenance of early stages of spermatogenesis, including supporting proper contacts between cyst cells and germ cells during cyst formation and preventing GSC tumorigenesis (Figure 6A). Despite the fact that we did not observe *integrin* transcripts among Belle mRNA targets in our CLIP-seq analysis (Appendix A), we speculate that Belle indirectly regulates unknown steps of the accumulation of integrin heterodimers on the surface of CySCs and early cyst cells, ensuring integrin complex localization or stability. There are at least three adaptor proteins that mediate the attachment of integrin complexes to the cellular actin cytoskeleton, including Talin (Rhea in *Drosophila*), spectraplakin Shot stop (Shot), and integrin-linked kinase (ILK) [72]. According to the GO analysis of the CLIP-seq data, factors responsible for actin filament organization and adherens junctions including Shot, are enriched among Belle mRNA targets (Appendix A). This observation elucidates a possible mode of regulation of integrin-based adhesion complexes in cyst cells; however, this requires further investigation.

Despite the restoration of early stages of spermatogenesis by the expression of a transgene copy of *β-integrin* in somatic cyst cells, these males were sterile owing to developmental disorders at the spermatid stage (Appendix A). It has been found in the screen study [54] that *RNAi*-induced knockdowns of *ter94* and *arf102f* in somatic cyst cells lead to spermatid defects. *ter94* and *arf102f* mRNAs were determined in our CLIP-seq assay as potential Belle targets with peaks in the 5′-UTR of *ter94* transcript and the coding region of *arf102f* (Appendix A). However, the involvement of these factors in the development of spermatids requires further study.

Our data indicate different functional consequences of *belKDs* in the germline and in somatic cyst cells of the testes (Figure 6B). Whereas *belKD* in the germline leads to a total germ cell loss with preservation of somatic testis cells as described previously [25], *belKD* in cyst cells induces a generation of tumor-like clusters of early germ cells which actively proliferate, but are not capable to further differentiation (Figure 1D–E). Thus, we found that Belle functions in the testes were required cell-autonomously for the maintenance of early germ cells including GSCs and non-autonomously in cyst cells to ensure germ cell differentiation. This dual role of Belle appears to reflect the diverse functions of human orthologs DDX3X and DBY in the soma and germline of the human testes.

Being involved in a wide range of cellular processes, DEAD-box RNA helicases have proved to be challenging to study. The interaction of DEAD-box RNA helicases with their mRNA targets is considered non-processive and transient [1,3]. Here we used the powerful CLIP-seq approach based on in vivo RNA-protein crosslinking to firstly identify a range of potential targets of Belle regulation in the testes. CLIP-seq analysis revealed that Belle interacts with more than 300 transcripts of protein-coding genes in the testes (Figure 4, Appendix A). According to the CLIP-seq data, we revealed the binding of Belle with transcripts of protein-coding genes both in 5′- and 3′-UTRs and the coding regions (Figure 4E, Appendix A). This indicates that Belle interacts predominantly with mature mRNAs and preferentially acts as a translational regulator. In accordance with previously published data, Belle/DDX3 can be involved both in translational repression and translational activation of target transcripts [18,26,27,28,73,74,75,76].

Our CLIP-seq analysis provides entry points in attempting to understand the molecular mechanisms underlying the spermatogenesis defects observed in both in somatic cyst cells and in the germline with failures of Belle expression. Using comparative analysis of the recently published screens, we selected a subset of genes that overlap with the ones detected in our CLIP-seq analysis (Appendix A). These screens focused on the identification of genes that are essential in cyst cells or in the germline of *Drosophila* testis during specific stages of spermatogenesis [53,54,55]. Comparative analysis revealed that *RNAi* knockdowns of 17 genes, which were determined in the CLIP-seq analysis as potential Belle targets, led to shared testis abnormalities with *belKDs* in cyst cells (tumor-like clusters of early germ cells) or in the germline (germ cell loss) (Figure 5A; Appendix A). Among them we selected four genes, *not1, rpn7, rho1,* and *caf1-55*, knockdowns of which provide the same morphological defects in the testes, according to data of all the three independent screens (Figure 5A). We revealed a decreasing level of Not1 and Caf1-55 in the testes with *RNAi belKD* in cyst cells (Figure 5B). We also experimentally confirmed that *RNAi not1KD* (Figure 5C–E) and *caf1-55KD* (Appendix A) in cyst cells can lead to the generation of the tumor-like germ cell clusters. Not1 is a key component of the CCR4-Not deadenylation complex involved in translational regulation of multiple mRNAs in development [28,77]. Caf1-55 is a component of the chromatin assembly factor 1 (CAF-1) complex, and also the nucleosome remodeling and deacetylase (NuRD) and polycomb repressive complex 2 (PRC2) chromatin complexes; it is essential for control of GSC self-renewal and proliferation both in the ovaries and testes [78,79]. We suggest that Belle positively regulates translation of *not1* and *caf1-55* mRNAs both in somatic cyst cells and germ cells. We verified the *RNAi* knockdown phenotypes for two other genes of the set, *rpn7* and *Fs(2)Ket*. Rpn7 is involved in the proteasome-mediated ubiquitin-dependent protein catabolic process [55], whereas β-importin Fs(2)Ket is responsible for the nuclear import and export [80]. We found that both *rpn7KD* and *Fs(2)KetKD* in cyst cells caused germline cluster formation with reproducible but relatively low frequency (Appendix A). We revealed that germline-specific *RNAi* knockdowns of at least four Belle target candidates, *not1, caf1-55, rpn7,* and *Fs(2)Ket,* led to a total loss of testis germ cells. Interestingly, in the cases of knockdowns of *caf1-55, rpn7,* and *Fs(2)Ket*, we observed the maintenance of somatic cells in the testes, whereas *RNAi* knockdown of *not1* caused a premature decrease both in somatic and germ cells (Appendix A). Note that failures of expression of Caf1-55, Rpn7, and Fs(2)Ket proteins in germ cells accurately reproduced the corresponding cell-autonomous maintenance defects observed in the most testes of males with the germline-specific *belKD* according the our previous study [25].

According to the screen data and our results, knockdowns of several selected Belle targets lead to similar developmental disorders in the testes (Figure 5A,D,E; Appendix A, Figure 6B). Assuming that Belle is an upstream translational regulator of these transcripts, we propose that failures of translation of a number of mRNA targets could cooperatively contribute to the morphological and developmental testis defects observed in *belKDs* both in somatic cyst cells and the germline (Figure 6B). Comparison of this subset with the data of CLIP-seq analysis of human DDX3 mRNA targets in HEK293T cells [75] revealed transcripts of orthologous human genes, including the *not1*, *caf1-55, hyd,* and *beta’COP* orthologs *Cnot1*, *RBBP4, UBR5,* and *COPB1* (Appendix A).

Although we found that Belle facilitated translation of *not1* and *caf1-55* mRNAs (Figure 5B), it appears to be a negative regulator of translation of *gbb* transcript in cyst cells (Appendix A). The hub and CySCs are known to secrete two BMP ligands, Gbb and Dpp, in the wild-type testes. Gbb appears to be more important for GSC maintenance in the testes, while Dpp is more significant in the ovarian niche [63,64]. This short-range signaling accomplished by these ligands is critical for the maintenance and self-renewal of GSCs. Diffusion of BMP ligands in wild-type testes is tightly restricted by a one-cell distance from the niche. However, it is shown that ectopic overexpression of BMP ligands itself does not cause abnormal expansion of GSCs in the testes [63,64]. Accordingly, we found no significant changes in the level of the key germline component of BMP cascade pSmad in the *belKD* testes (Appendix A). So, the increased concentration of Gbb secreted by cyst cells into the internal testis space is unlikely to be the main causative factor of the formation of GSC-like clusters of germ cells. We speculate that Belle can normally negatively regulate *gbb* translation in differentiating cyst cells, preventing inappropriate activation of BMP signaling outside the niche.

In our previous paper we demonstrated that Belle is required cell-autonomously for mitotic progression and survival of GSCs and spermatogonia as the upstream regulator of mitotic cyclin expression [25]. A disruption of Belle expression leads to a strong germ cell loss in the testes; the phenotype mimics that of SCOS in humans with *DDX3Y* deletion [25,29]. Here we dissected a non-autonomous role of Belle (DDX3) in the prevention of tumorigenesis of early germ cells. To date, DDX3 helicase has attracted increasing attention owing to its essential but contradictory functions in the progression of human cancers in different tissues. The role of DDX3 as an oncogene or tumor suppressor is associated with activity in the tissue of predominant signaling pathways in which DDX3 is involved [11,12,13]. Our data augment and enrich our current understanding of the testis-specific functions of Belle/DDX3 as an essential translational regulator. Considering that somatic cyst cells of *Drosophila* testes appear to be the functional analogs of Sertoli cells of mammalian testes [34], the employment of the *Drosophila* model may contribute to the elucidation of DDX3 functions in the maintenance of spermatogenesis and shed light on tumorigenesis and cancer treatment in the human testes.

## Figures and Tables

**Figure 1 cells-09-00550-f001:**
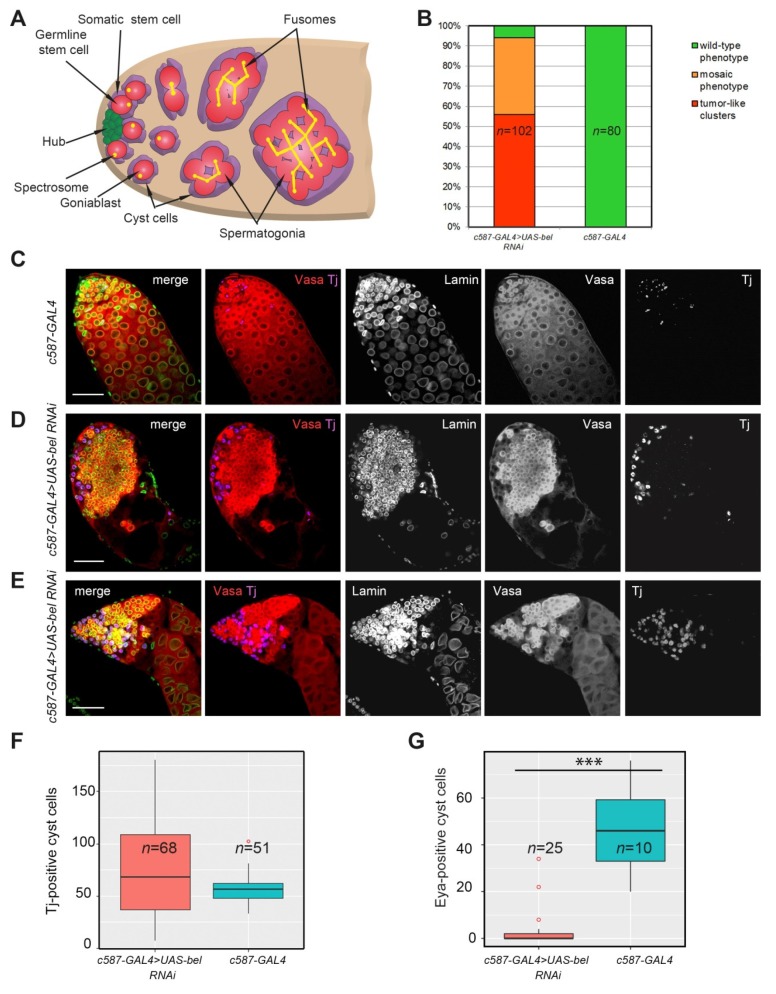
Phenotypic analysis of the testes of males with *RNAi belKD* in somatic cyst cells and the control *c587-GAL4* driver line. (**A**) Scheme of the apical tip of *Drosophila* testis. Germline stem cells (red) are located near the hub (green) and are encapsulated by two somatic cyst stem cells (violet). A daughter of a germline stem cell (GSC), the goniablast (red), undergoes four cell divisions to generate cysts of 2–16 spermatogonia. The 16-cell cyst enters the spermatocyte growth program and meiosis (not shown). Dot-like structures in GSCs and goniablasts, spectrosomes (yellow dots), become branched fusomes in the secondary spermatogonial cells and spermatocytes. (**B**) *belKD* in cyst cells leads to the appearance of tumor-like germ cell clusters. The bar diagram depicts the distribution of testis phenotypes of *c587-GAL4>UAS-bel RNAi* and control *c587-GAL4* 0–1 day-old males: wild-type (green), mosaic (orange), and strong tumor-like phenotype (red). (**C–E**) Testes of newly eclosed males with *belKD* in cyst cells and of control males were immunofluorescently stained with anti-Vasa (red), anti-Lamin (green), and anti-Traffic jam (Tj, violet) antibodies. Lamin staining indicates nuclear envelope positions. Internal confocal slices of the immunostained whole-mount fixed testis preparations are shown. The apical tips of the testes are oriented leftward on all panels here and after. Scale bars are 50 µm. (**C**) Testes of the control *c587-GAL4* line represented the wild-type phenotype. (**D**) More than one half of the analyzed *c587-GAL4 > UAS-bel RNAi* testes (55.9%) contained tumor-like clusters of early germ cells with Tj-stained somatic cyst cells found to be segregated from germ cells. (**E**) Another fraction of *c587-GAL4>UAS-bel RNAi* testes represented a mosaic phenotype and contained cysts of spermatogonia and spermatocytes along with tumor-like clusters. (**F**) Tj-stained cyst cells are insignificantly increased in number in the testes upon cell-autonomous *belKD* in cyst cells. The boxplot diagram shows the number of Tj-positive early cyst cells per testis in *c587-GAL4>UAS-bel RNAi* and control *c587-GAL4* testes. Statistical analysis was performed by Wilcoxon/Mann–Whitney two-sided test, *p* = 0.073. (**G)** Eya-stained cyst cells are lost upon *belKD* in the cyst cells. The boxplot diagram shows the number of Eya-positive late cyst cells per testis in *c587-GAL4>UAS-bel RNAi* and control *c587-GAL4* testes. Statistical significance (***) was determined by Wilcoxon/Mann–Whitney two-sided test, *p *= 4.528∙× 10^−6^. The bold lines on the boxes (**F, G**) mean median values.

**Figure 2 cells-09-00550-f002:**
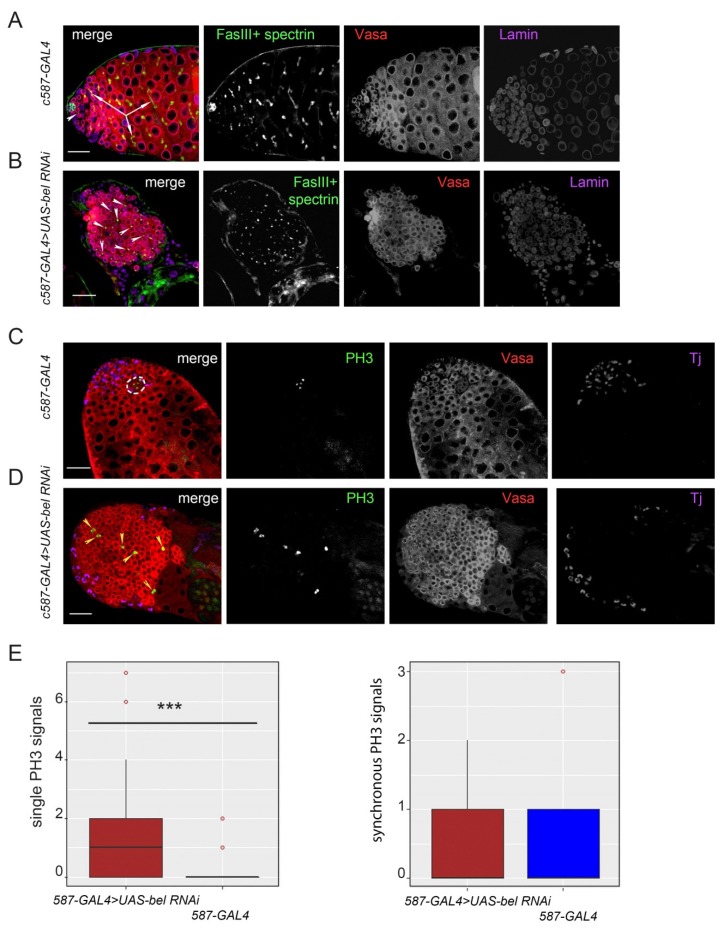
Analysis of germ cell proliferation activity in the clusters. (**A, B)** Testes of newly eclosed males were stained with anti-Vasa (red), anti-Lamin (violet), anti-Fasciclin III (FasIII) (green), and anti-α-spectrin (green) antibodies. The Lamin staining indicates nuclear envelope positions. Internal confocal slices of the immunostained testis preparations are shown. Scale bars are 30 µm. (**A**) Control testes of *c587-GAL4* males display a wild-type pattern of spectrosomes (white arrowheads) and fusomes (white arrows). A black asterisk marks the FasIII-stained hub. (**B**) Germ cells in the clusters of the *RNAi belKD* testes contain dot-like spectrosomes (white arrowheads). (**C**, **D**) Testes of newly eclosed males were stained with anti-Vasa (red), Tj (violet), and anti-PH3 (green) antibodies. Internal confocal slices are shown. Scale bars are 30 µm. (**C**) Spermatogonial cells in the cysts of the control testes undergo synchronous divisions (outlined region). (**D**) Germ cells in the clusters of *RNAi belKD* testes divide independently of each other and asynchronously (yellow arrowheads). (**E**) Boxplot diagrams show the numbers of PH3 signals in germ cells per testis calculated for the testes of *c587-GAL4>UAS-bel RNAi* flies (*n* = 20) and control *c587-GAL4* testes (*n* = 18). Statistically significant differences (***) were found between the two groups for single signals (left diagram) by Wilcoxon/Mann–Whitney two-sided test, *p* = 0.0093. For synchronous signals in germ cells (right diagram) no significant differences were found between the experimental and control groups (*p* = 0.8886).

**Figure 3 cells-09-00550-f003:**
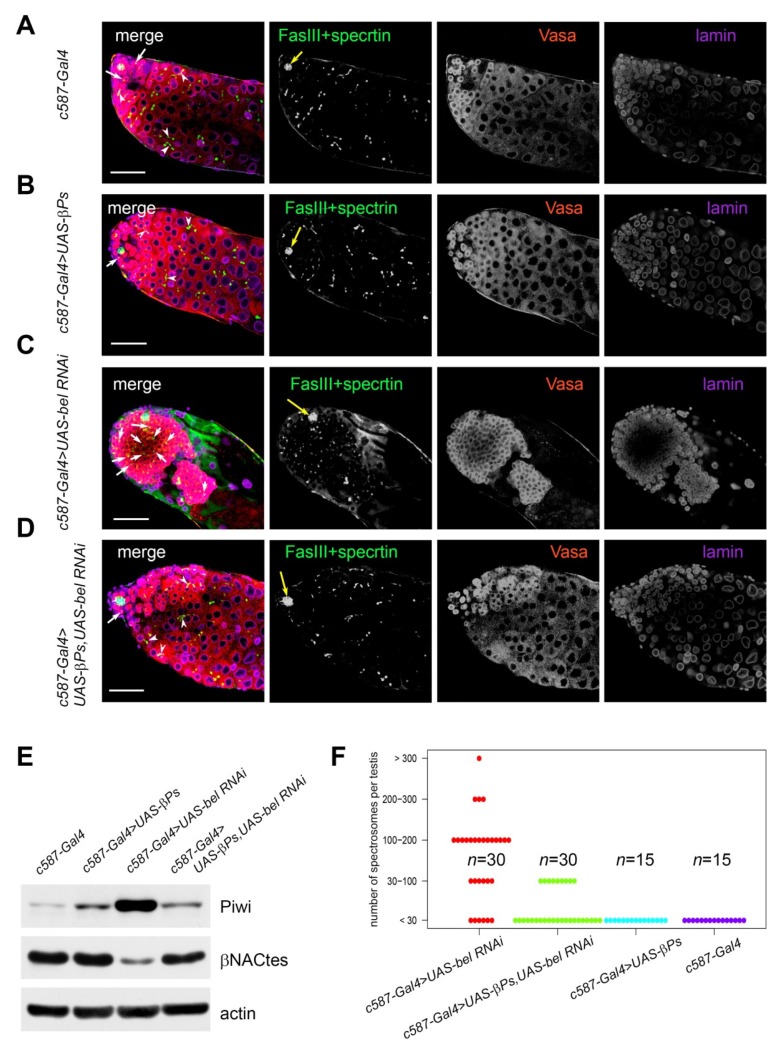
Expression of a transgenic copy of *β-integrin* in cyst cells rescued early stages of spermatogenesis in the *RNAi belKD* testes. (**A**–**D**) Testes of young males were stained with anti-Vasa (red), anti-Fasciclin III (green), anti-α-spectrin (green), and anti-Lamin (violet) antibodies. Yellow arrows indicate hub positions. White arrows indicate spectrosomes, white arrowheads indicates fusomes. Scale bars are 30 µm. (**A**) The testes of *c587-GAL4* flies were used as a wild-type control. (**B**) The testes of *c587-GAL4>UAS-βPS* flies did not exhibit defects of germ cell differentiation. (**C**) The testes of *c587-GAL4>UAS-bel RNAi* males contained clusters of undifferentiated early germ cells. (**D**) Ectopic expression of a *βPS (β-integrin)* copy in cyst cells in the background of *RNAi belKD* (*c587-GAL4>UAS-βPS; UAS-bel RNAi)* led to the restoration of premeiotic stages of spermatogenesis. (**E**) Western blot analysis of testis lysates. The marker of germline stem cells Piwi returned to a wild-type level in the testes of *c587-GAL4>UAS-βPS; UAS-bel RNAi* males. The marker of spermatocytes βNACtes was restored to its wild-type level in the testes of *c587-GAL4>UAS-βPS; UAS-bel RNAi* males. Loading was 45 µg per lane. Anti-actin antibodies were used as a loading control. (**F**) Counts of spectrosomes in the testes of the experimental and control males. In most of the experimental testes (*belKD + β-int*) the number of spectrosomes did not exceed 30 as in wild-type control.

**Figure 4 cells-09-00550-f004:**
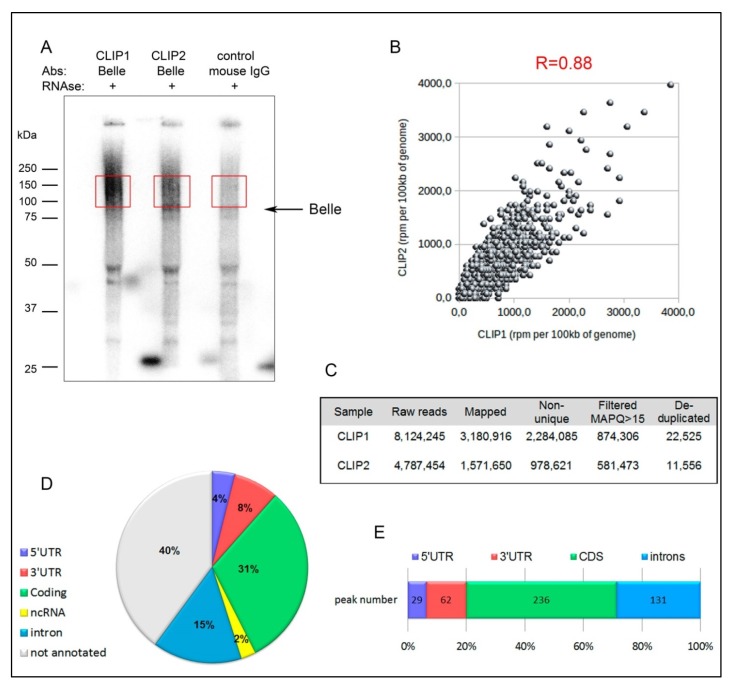
Transcriptome-wide identification of Belle mRNA targets by CLIP-seq analysis. (**A**) Autoradiogram of ^32^P-labeled cross-linked RNA-Belle complexes. Two independent immunoprecipitations using anti-Belle antibodies and UV cross-linked testes and a control experiment with non-immune mouse IgG were performed. The recovery of cross-linked target RNAs in RNA-Belle complexes was monitored by autoradiography after RNAse A treatment and 5′-end labeling with γ^32^P fragments of the membrane containing the main radioactive signals above 85 kDa (arrow indicates position non-crosslinked Belle) that are outlined by red squares were cut for further library processing. (**B**) A scatterplot of rpm of uniquely mapped reads per 100 kb for the CLIP 1 and CLIP 2 experiments. Pearson’s correlation coefficient is used to characterize the reproducibility of two CLIP-seq experiments. Each dot represents rpm per 100 kb of genome. (**C**) Table summarizing the number of counting reads for the two independent Belle CLIP-seq experiments (CLIP1 and CLIP2). (**D**) Distribution of uniquely mapped peaks across the whole genome (non-annotated genomic regions; non-coding RNAs; 5′-UTRs; 3′-UTRs; CDSs, introns). (**E**) Distribution of CLIP peaks mapped to protein-coding gene regions.

**Figure 5 cells-09-00550-f005:**
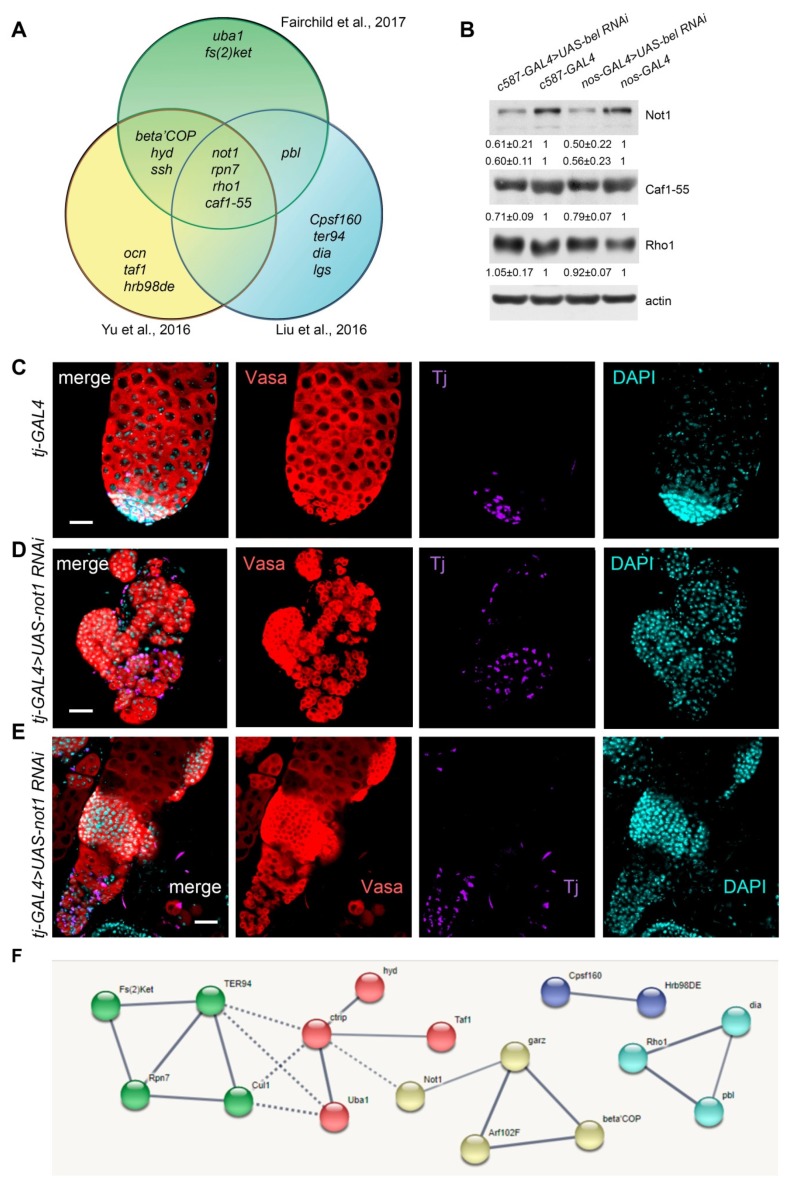
Comparative analysis of candidate targets of Belle CLIP-seq with the results of published screens. (**A**) A Venn diagram shows the overlap of genes, *RNAi* knockdowns of which provide the morphological defects similar to *RNAi belKD* in the testes according to data of three independent screens [53,54,55], and which were determined in our CLIP-seq analysis as putative mRNA targets of Belle. Knockdowns of four genes, *not1, rpn7, rho1,* and *caf1-55*, provide the same morphological defects in the testes, according to data from all three independent screens. (**B**) Western blot analysis of testis lysates of *c587-GAL4>UAS-bel RNAi* males, *nos-GAL4>UAS-bel RNAi* males, and the corresponding driver lines as controls. Anti-actin antibodies were used as a loading control. The values (obtained by using ImageJ software) were normalized to the loading control level (actin) and then to the values obtained for the control lysates. Mean values and standard errors are presented for at least three independent experiments. We revealed a decrease in the level of Not1 and Caf1-55 proteins in the testes with *belKDs*; however, we did not detect significant changes in the level of Rho1. (**C**–**E**) Immunostaining of fixed testis preparations of males with *RNAi* knockdown of *not1* using soma-specific driver *tj-GAL4*. Testes of newly eclosed males were stained with anti-Vasa (red) and anti-Tj (violet) antibodies; chromatin was stained by DAPI (4′,6-diamidino-2-phenylindole, blue). Scale bars are 30 µm. (**C**) Control testes contained a wild-type array of cells at all stages of spermatogenesis. (**D**) Most of *not1KD* testes are filled by tumor-like clusters of small germ cells and somatic cyst cells placed separately. (**E**) In the remaining *not1KD* testes clusters of small germ cells were found along with differentiating germ cells. (**F**) Interaction cluster analysis of genes from the full list of Belle CLIP-targets shared with the results of the screen studies (Appendix A) was performed by STRING database tools. Only 17 of 22 genes form five interacting clusters with high confidence settings (interaction score 0.7). The thickness of the line between genes (color balls) reflects the relative probability of interaction. See Appendix A for a detailed description.

**Figure 6 cells-09-00550-f006:**
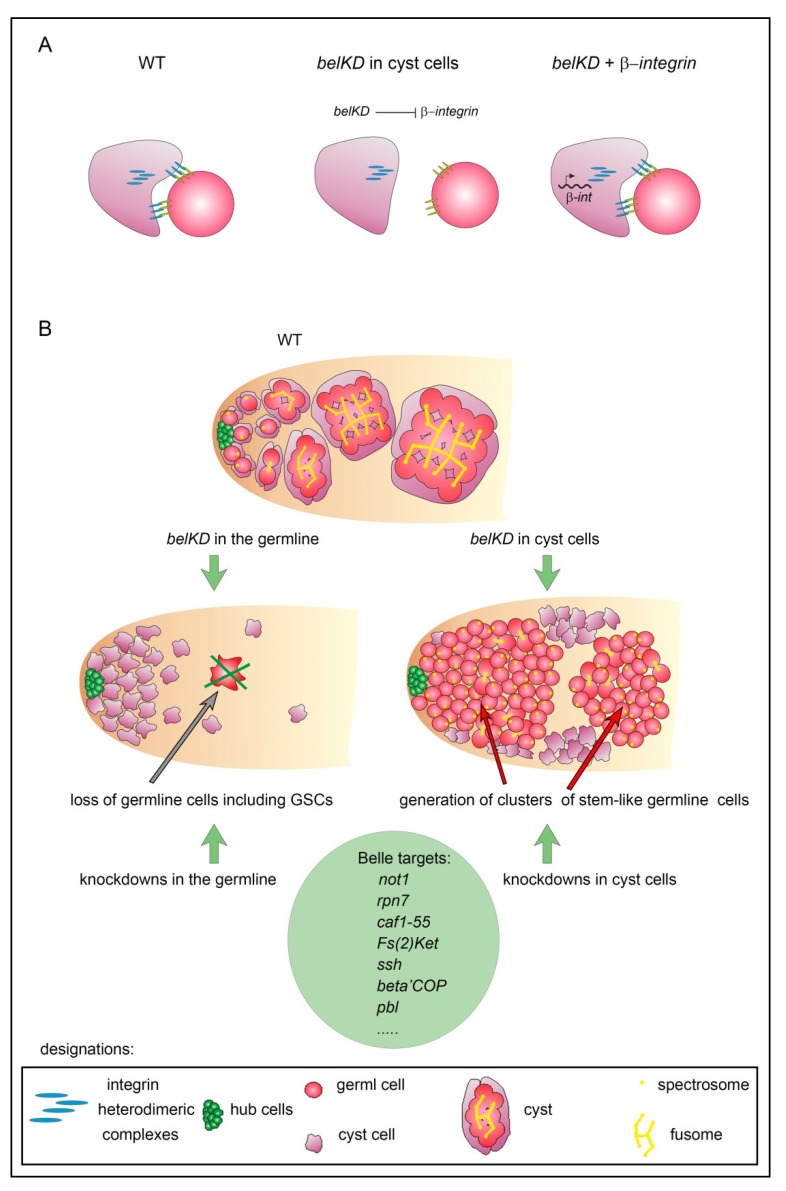
Belle functions in cyst cells ensuring spermatogenesis maintenance. (**A**) A model of how Belle regulates cyst formation and prevents tumorigenesis of early germ cells in the *Drosophila* testis via indirect control of integrin adhesion complexes, ensuring localization or stability of integrin complexes. Transgenic overexpression of *β-integrin* in cyst cells provided restoration of premeiotic stages of spermatogenesis in the *RNAi belKD* testes. (**B**) Dual role of Belle in spermatogenesis. Representation of the different defects of early stages of spermatogenesis caused by *RNAi belKDs* in the germline and somatic cyst cells. Whereas *belKD* in the germline leads to total germ cell loss with the maintenance of somatic cyst cells [25], *belKD* in cyst cells non-autonomously induces the generation of tumor-like clusters of early germ cells. Below: knockdowns of selected Belle targets lead to the same developmental disorders in the testes. Designations for A and B are presented in the box at the bottom.

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
