# Peer review of "The Drosophila RNA Helicase Belle (DDX3) Non-Autonomously Suppresses Germline Tumorigenesis Via Regulation of a Specific mRNA Set"

_cells, 2020, doi:10.3390/cells9030550_

Round 1

Reviewer 1 Report

The manuscript by Kotov et al. analyzes the function of the RNA helicase Belle in somatic cyst cells of Drosophila testes. The authors have previously shown that belle (bel) KD in germ cells in testes lead to germ cell loss. Here they show that bel KD in somatic cyst cells that surround germline stem cells (GSCs) lead to a defect in GSC differentiation and the formation of GSC tumors. Then, they try to address the role of adhesion molecules in this phenotype and show that they can rescue the GSC differentiation defect following overexpression of beta-integrin. They identify a number of Bel mRNA targets through CLIP assays in testes. Finally, using western blots, they record protein levels of a number of genes that are present or not in the CLIP data sets to molecularly link the phenotypic defects to bel mRNA targets. The paper does not present functional data to clearly identify the role of specific mRNA targets in Bel function. Despite this point, the paper reports interesting data. The experiments are well controlled (more so in the first part of the manuscript). I have a number of concerns and comments that should be addressed before publication.

Major comments:

1) In Figure 2C, the outline showing synchronous divisions is not visible. In Figure 2E, the frequency of synchronous divisions is quantified to be similar in the wild type and bel KD. How are synchronous divisions recorded in the context of undifferentiated germ cell tumors (bel KD)?

2) The title of Paragraph 3.3 (line 489) indicates that bel KD leads to a defect in cell adhesion. However, data in this paragraph do not show this, since all experiments gave negative results. This title should be changed.

3) Controls are required to conclude about the non-rescue with expression of UAS-armS10 and the rescue with expression of beta-integrin. In both cases, the rescue transgene is an UAS transgene, therefore Gal4 expressed from c587-Gal4 must activate two transgenes (UAS-bel RNAi and the rescue transgene). It is possible that UAS-armS10 cannot be activated by Gal4 due to a position effect in the genome. On the other hand, the rescue with beta-integrin could result from a lower expression of UAS-bel RNAi in this context, due to the recruitment of Gal4 to the UAS-beta-integrin transgene. In both rescue conditions (armS10/ beta-integrin), bel levels should be quantified, as well as expression of the transgenes (using immunostaining or RT-qPCR).

4) In the presence of UAS-beta-integrin expressed in somatic cells (with c587-Gal4), interactions and signaling between somatic and germ cells are proposed to be restored, and germ cells differentiate more or less normally. However germ cells are affected at a late stage (spermatid) and these males are sterile. Could the authors speculate which kind of pathways might be affected, which would lead to such a late defect in germ cells following bel KD in early somatic cells?

5) For the CLIP assays in Figure 4, a control without UV (if available) would be useful.

6) The text p. 23 indicates that CLIP reads are enriched in 5' and 3' UTRs and comparatively less enriched in introns. However, Figure 4E shows that 20% of CLIPs are in 5' and 3' UTRs whereas 28.6% are in introns. Are these enrichments relative to the size of UTRs and introns in the whole genome? Binding to introns could indicate a role in splicing/alternative splicing.

7) I find the part of the manuscript following CLIP assays confusing.

First, it is indicated on line 633 that 4 potential Bel targets were selected, however, there are no data on one of them: rpn7.

Second, these 4 genes were selected because their KD shows similar defects as bel KD. Still, the authors go on confirming this information by analyzing the defects following KD of one of these 4 genes, namely not1. Given that this information is already available from published studies, if the purpose is to compare the defects with those of Bel KD, the same Gal4 driver, c587-Gal4, should be used. Experiments with the germline driver nos-Gal4 would also be useful.

Third, this experiment does not clarify the relationships between bel and not1. A better experiment to address this point would be to try to rescue bel KD defects by overexpressing not1.

Fourth, the same applies to caf1-55. This information (line 680) should be removed, or the data should be shown.

8) The rationale in the two last paragraphs on p. 28 is not clear. Why Gbb/Dpp are separated from the rest of signaling pathways? It might be clearer if all signaling pathways were described in a single paragraph that would also indicate which mRNAs within these pathways are in the CLIP data sets. This would help understanding which regulation might be direct.

9) The meaning of the sentence starting: "We often observed... ", line 703 is unclear to me. It is contradictory.

10) Adhesion between somatic and germ cells was not analyzed per se. Therefore, the sentence starting "We determined that ...", line 733 is overstated.

11) The model proposing that an important defect is the lack of integrin localization at the surface of cyst cells (Figure 6) should be validated using immunostaining.

12) Figures 6 and 7 are very big and redundant. They should be replaced by a single graphic integrating the different data and information.

Minor points:

Line 110: and enter meiosis resulting in the generation .....

Line 339: for only one CLIP library and present in the second CLIP library ...

Line 438: displayed a wild-type pattern of ...

Line 505: formation of cadherin-catenin adhesion ...

Line 688: , and RNA processing factors were identified (Table S4).

Line 691: in the testes of males with belKD ...

Line 827: We suggest that ...

Line 831: Assuming that Belle is an upstream ...

Line 844: accomplished by these ligands is critical ...

Author Response

Author's Reply to the Review Reports (Reviewer 1)

Reviewer 1

 Comments and Suggestions for Authors

The manuscript by Kotov et al. analyzes the function of the RNA helicase Belle in somatic cyst cells of Drosophila testes. The authors have previously shown that belle (bel) KD in germ cells in testes lead to germ cell loss. Here they show that bel KD in somatic cyst cells that surround germline stem cells (GSCs) lead to a defect in GSC differentiation and the formation of GSC tumors. Then, they try to address the role of adhesion molecules in this phenotype and show that they can rescue the GSC differentiation defect following overexpression of beta-integrin. They identify a number of Bel mRNA targets through CLIP assays in testes. Finally, using western blots, they record protein levels of a number of genes that are present or not in the CLIP data sets to molecularly link the phenotypic defects to bel mRNA targets. The paper does not present functional data to clearly identify the role of specific mRNA targets in Bel function. Despite this point, the paper reports interesting data. The experiments are well controlled (more so in the first part of the manuscript). I have a number of concerns and comments that should be addressed before publication.

Thank you for your very helpful comments. We have incorporated our responses in the revised version of the paper, which has clearly improved the manuscript.

Major comments:

1) In Figure 2C, the outline showing synchronous divisions is not visible. In Figure 2E, the frequency of synchronous divisions is quantified to be similar in the wild type and bel KD. How are synchronous divisions recorded in the context of undifferentiated germ cell tumors (bel KD)?

In Figure 2C the missed outline has been restored.

According to our data about 40% of the RNAi belKD testes have mosaic phenotype, see lines 375-378 of the main text “Another fraction of the RNAi belKD testes (38.2%, 39 of 102 cases) exhibited the so-called mosaic phenotype with several small germ cell clusters in the neighborhood with spermatogonial cells and spermatocytes (Figure 1B,E)”. Synchronous divisions of spermatogonial cells were recorded in the experimental testes which exhibited the mosaic phenotype. Note that the number of the synchronous signals (recorded as a single event) per unit of time was rather not numerous and most often such signals were absent both in the experimental and in the control testes (Figure 2E).

2) The title of Paragraph 3.3 (line 489) indicates that bel KD leads to a defect in cell adhesion. However, data in this paragraph do not show this, since all experiments gave negative results. This title should be changed.

It was changed accordingly. The new title of Paragraph 3.3 is “Ectopic overexpression of a transgenic armS10 copy in cyst cells did not restore Belle knockdown phenotype in the testes”.

3) Controls are required to conclude about the non-rescue with expression of UAS-armS10 and the rescue with expression of beta-integrin. In both cases, the rescue transgene is an UAS transgene, therefore Gal4 expressed from c587-Gal4 must activate two transgenes (UAS-bel RNAi and the rescue transgene). It is possible that UAS-armS10 cannot be activated by Gal4 due to a position effect in the genome. On the other hand, the rescue with beta-integrin could result from a lower expression of UAS-bel RNAi in this context, due to the recruitment of Gal4 to the UAS-beta-integrin transgene. In both rescue conditions (armS10/ beta-integrin), bel levels should be quantified, as well as expression of the transgenes (using immunostaining or RT-qPCR).

We agree with the reviewer in that RNAi-mediated knockdowns can cause a reduction of gene expression level to a variable degree and in some cases, it can result in a negligible effect. It is possible to quantify the level of transgene expression using cells of the same type, for example, cell culture. However, Drosophila testes consist of multiple populations of different somatic and germ cells, with different cell numbers (Figure 1A). Note that the number of cyst cells per testis is significantly less than the number of germ cells. Since belle is normally expressed ubiquitously, it is very problematic to quantify the drop in belle expression in the case of knockdown only in cyst cell population by Western blot, RT-qPCR or immunostaining (the last one is a non-quantitative method, and in our case belKD leads to significant violation of whole testis morphology and to change in the shape of cyst cells from a flat elongated shape to a spherical one). We should note, however, that UAS-bel RNAi line (VDRC #6299) has the proven effectiveness and widespread using (Dietzl et al., 2007, Nature; Poulton et al., 2011; Development; Kotov et al., 2016, Mol Biol Cell; Mummery-Widmer et al., 2009, Nature; Lee et al., 2016, Cell; Cusmano et al., 2019, Front. Physiol, etc.). To obtain evidence of driver-activated expression of the transgenes we performed Western blot analysis using anti-Arm and anti-beta-integrin antibodies and found that in both rescue experiments the levels of these proteins were increased. We included these data in Supplementary material (Fig. S4C and Fig. S5A). Note that the expression of transgenic Arm copy did not lead to a restoration of early stages of spermatogenesis in the background of belKD in cyst cells, but the expression of beta-integrin did. In addition, although we cannot quantify the level of bel knockdown, we have shown that male fertility was not restored by over-expression of beta-integrin, indicating the existence of other disorders caused by belKD in somatic cyst cells (Fig. S5B,C).

4) In the presence of UAS-beta-integrin expressed in somatic cells (with c587-Gal4), interactions and signaling between somatic and germ cells are proposed to be restored, and germ cells differentiate more or less normally. However germ cells are affected at a late stage (spermatid) and these males are sterile. Could the authors speculate which kind of pathways might be affected, which would lead to such a late defect in germ cells following bel KD in early somatic cells?

Yes, we propose that a failure in the translation of candidate Belle targets could lead to post-meiotic disorders of spermatid development. Using DAVID (Functional Annotation Chart DAVID v6.8) we found at least 5 genes among CLIP-detected Belle targets, which are included in category GO:0007291~sperm individualization (see Table S2, list “Sum GO chart graph”). There are Past1, orb2, heph (PTB), cul3, and ntc. However, among them orb2, Past1, ntc and heph exhibit a germline-specific expression in the testes (Xu et al., 2014; Olswang-Kutz et al., 2009; Bader et al., 2010; Robida and Singh, 2003). Cul3-based enzyme complex is required for caspase activation during spermatid individualization, however, differential expression of cul3 germline-specific and soma-specific transcripts are found to account for the distinct phenotypes (male sterility in the germline versus lethality in somatic cells) (Arama et al., 2007).

It has been found in the screen study that RNAi-induced knockdowns of ter94 and arf102f in somatic cyst cells lead to spermatid defects (Fairchild et al., 2016). ter94 and arf102f mRNAs were determined in our CLIP assay as potential Belle targets with peak regions in the 5’UTR of ter94 transcript and in the coding region of arf102f (Figure S6B; Table S1 and Table S3). However, this issue requires further investigation. We have introduced the corresponding explanations in the Discussion part of the main text.

Cited references:

Xu S, Tyagi S, Schedl P. Spermatid cyst polarization in Drosophila depends upon apkc and the CPEB family translational regulator orb2. PLoS Genet. 2014 10(5):e1004380. doi: 10.1371/journal.pgen.1004380.

Hempel LU, Rathke C, Raja SJ, Renkawitz-Pohl R. In Drosophila, don juan and don juan like encode proteins of the spermatid nucleus and the flagellum and both are regulated at the transcriptional level by the TAF II80 cannonball while translational repression is achieved by distinct elements. Dev Dyn. 2006 Apr;235(4):1053-64.

Olswang-Kutz Y, Gertel Y, Benjamin S, Sela O, Pekar O, Arama E, Steller H,Horowitz M, Segal D. Drosophila Past1 is involved in endocytosis and is required for germline development and survival of the adult fly. J Cell Sci. 2009 Feb15;122(Pt 4):471-80. doi: 10.1242/jcs.038521.

Robida MD, Singh R. Drosophila polypyrimidine-tract binding protein (PTB) functions specifically in the male germline. EMBO J. 2003 Jun 16;22(12):2924-33. PubMed PMID: 12805208; PubMed Central PMCID: PMC162153.

Arama E, Bader M, Rieckhof GE, Steller H. A ubiquitin ligase complex regulates caspase activation during sperm differentiation in Drosophila. PLoS Biol. 2007 Oct;5(10):e251.

Bader M, Arama E, Steller H. A novel F-box protein is required for caspase activation during cellular remodeling in Drosophila. Development. 2010 May;137(10):1679-88. doi: 10.1242/dev.050088.

5) For the CLIP assays in Figure 4, a control without UV (if available) would be useful.

Control without UV cross-linking is not generally used in CLIP assay because of a relatively long time period that is required for technical manipulations with the immunoprecipitated RNP complex (See Materials and Methods section “CLIP-seq analysis”). It is practically impossible to maintain stable RNA-protein interaction without covalent cross-linking for several days, which are necessary to perform the protocol steps according to the CLIP method. It is also known that the interaction of DEAD-box RNA helicases with their mRNA targets is a very transient process and ATP hydrolysis leads to rapid helicase release from the corresponding RNA. For immunoprecipitation of DEAD-box RNA helicase-mRNA complexes covalent crosslinking between the protein and RNA targets is a necessary condition.

6) The text p. 23 indicates that CLIP reads are enriched in 5' and 3' UTRs and comparatively less enriched in introns. However, Figure 4E shows that 20% of CLIPs are in 5' and 3' UTRs whereas 28.6% are in introns. Are these enrichments relative to the size of UTRs and introns in the whole genome? Binding to introns could indicate a role in splicing/alternative splicing.

We write that “Across transcripts of protein-coding genes Belle was found to be enriched in coding sequences and both in 5′- and 3′-UTRs, and exhibited comparatively low binding to introns, 28.6% (Figure 4E)”. Actually, the average intronic size exceeds exonic size more than two-fold in the Drosophila melanogaster genome (Hong et al., 2006). But we did not normalize data of peak enrichment on the relative size of intron and exon regions of genes. The higher peak enrichment in exonic regions compared to introns (71% versus 29%) suggests that Belle primarily binds to mature mRNAs. However, according to the previously published data, DDX3 helicases can be involved in a wide range of intracellular processes associated with RNA metabolism, including transcription and splicing (Kotov et al., 2014). Thus, we cannot exclude the participation of Belle in mRNA processing in the nucleus. We partially incorporated our observations in section 3.5 of the manuscript.

Hong X, Scofield DG, Lynch M. Intron size, abundance, and distribution within untranslated regions of genes. Mol Biol Evol. 2006 Dec;23(12):2392-404.

7) I find the part of the manuscript following CLIP assays confusing.

First, it is indicated on line 633 that 4 potential Bel targets were selected, however, there are no data on one of them: rpn7.

Unfortunately, we didn’t find antibodies to Rpn7. But we performed additional immunostaining analysis of the testis to study phenotype in the case of RNAi-induced rpn7 knockdowns in cyst cells and germ cells and introduced the results in the revised version of the manuscript (new Figure S8, new Figure S9). We also analyzed RNAi-induced knockdown of another candidate Belle target, beta-importin Fs(2)Ket (Figure 5A, Table S3), that is responsible for nuclear import and export. We found that both rpn7KD and Fs(2)KetKD in cyst cells caused germline cluster formation with reproducible but low frequency (new Figure S8), whereas germline-specific RNAi knockdowns of four Belle candidate targets, not1, caf1-55, rpn7, and Fs(2)Ket, led to total loss of testis germ cells (new Figure S9). According to our data knockdowns of rpn7 and Fs(2)Ket both in somatic cyst cells and in the germline led to similar phenotypes with corresponding belKDs (new Figure S8, new Figure S9).

Second, these 4 genes were selected because their KD shows similar defects as bel KD. Still, the authors go on confirming this information by analyzing the defects following KD of one of these 4 genes, namely not1. Given that this information is already available from published studies, if the purpose is to compare the defects with those of Bel KD, the same Gal4 driver, c587-Gal4, should be used.

Since we obtained a large number of candidate Belle targets as a result of the CLIP-seq analysis, we used a comparison with the results of screening studies to restrict the set of proteins, the absence of which leads to similar defects of spermatogenesis as in the case of belKD. Our subsequent plans include investigations of the influence of other Belle targets on the testis development and maintenance of spermatogenesis. In this manuscript, we focused only on several Belle targets. We have performed pilot experiments using the c587-Gal4 driver to confirm spermatogenesis defects in the case of KDs of the selected genes, not1 and caf1-55. However, we did not observe any differences from the wild-type testis phenotype in these experiments, possibly due to low knockdown efficiency. Using the tj-GAL4 driver supplemented by the expression of an additional dicer copy allowed to exhibit expected phenotypic defects in the testes with various frequencies. We included immunostaining data for knockdowns of caf1-55, rhoI, rpn7, and also Fs(2)Ket in somatic cells using the tj-GAL4 driver and in the germline using the nos-GAL4 driver in the revised manuscript (new Figure S8, new Figure S9).

Experiments with the germline driver nos-Gal4 would also be useful.

Yes, we also included immunostaining data for knockdowns of not1, caf1-55, rhoI, rpn7, and Fs(2)Ket in the germline using nos-GAL4 driver in the revised manuscript (new Figure S9) and corresponding discussion in the main text.

Third, this experiment does not clarify the relationships between bel and not1. A better experiment to address this point would be to try to rescue bel KD defects by overexpressing not1.

We believe that we have demonstrated the relationships between bel and not1 in several ways: 1) we identified that not1 mRNA directly bind to Bel in our CLIP-seq assay with peaks both in 5’UTR and CDS (Figure S6D, Table S1 and Table S3); 2) we found that belKD led to a decrease in Not1 protein level indicating positive translation regulation of Not1; 3) we showed that not1KDs in somatic cyst cells and in the germline are phenotypically similar to belKD that also supports bel and not1 relationship.

We agree that experiments with the rescue of belKD defects by overexpressing not1 would be useful. However, according to our hypothesis, the failures in the translation of a specific set of mRNA targets (but not only the not1 translation problem) cooperatively contribute to developmental defects observed in the testes with belKD (Figure 6B). In this case overexpression of not1 would not be enough for the rescue of belKD defects.

Fourth, the same applies to caf1-55. This information (line 680) should be removed, or the data should be shown.

Yes, we incorporated data for caf1-55KDs in the revised manuscript (new Figure S8, new Figure S9).

8) The rationale in the two last paragraphs on p. 28 is not clear. Why Gbb/Dpp are separated from the rest of the signaling pathways? It might be clearer if all signaling pathways were described in a single paragraph that would also indicate which mRNAs within these pathways are in the CLIP data sets. This would help in understanding which regulation might be direct.

We agree with the reviewer and fused together the two paragraphs in a single one. We also indicated Gbb as a Belle target on the Figure S10.

9) The meaning of the sentence starting: "We often observed... ", line 703 is unclear to me. It is contradictory.

Hub cells provide activation of the JAK/STAT signaling pathway by secreting the cytokine molecule unpaired (Upd) for supporting both types of testis stem cells, GSCs and CySCs (Kiger et al., 2001; Tulina and Matunis, 2001). GSCs are tightly adjusted to the hub in wild-type. However, GSC-like germ cells in the tumorous clusters do not maintain tight contacts with the hub and they appear not to receive short-range signals from the hub for their maintenance. According to the reviewer’s comment, we replaced in the main text the sentence “We often observed that germ cells in the clusters did not maintain tight contacts with the hub, indicating some independence of cluster maintenance from direct JAK-STAT signaling provided by the hub” by next one “We often observed that GSC-like germ cells in the clusters did not maintain tight contacts with the hub. Thus, the survival of germ cells in the clusters does not depend on the level of activation of the JAK-STAT pathway in them.”

10) Adhesion between somatic and germ cells was not analyzed per se. Therefore, the sentence starting "We determined that ...", line 733 is overstated.

We demonstrated the disruption of adhesion contacts between somatic and germ cells in 3.1 section of the results: “Using flies with the c587-GAL4-induced transgenic UAS-gfp construct along with UAS-bel RNAi we clearly established that GFP-marked somatic cyst cells mainly did not encapsulate early germ cells in the testes, but rather located separately from germ cell clusters being pushed to the testis shell (Figure S2A). These somatic cells had a round shape in contrast to their flat elongated shape in the control testes c587-GAL4>UAS-gfp (Figure S2B)”.

11) The model proposing that an important defect is the lack of integrin localization at the surface of cyst cells (Figure 6) should be validated using immunostaining.

We tried to do this, but unfortunately, antibodies to integrin practically do not work in immunostaining, it is problematic to distinguish a specific signal from the background.

12) Figures 6 and 7 are very big and redundant. They should be replaced by a single graphic integrating the different data and information.

Yes, we replace old Figures 6 and 7 by a single new Figure 6 according to request.

Minor points:

Line 110: and enter meiosis resulting in the generation .....

It was changed accordingly.

Line 339: for only one CLIP library and present in the second CLIP library ...

It was changed accordingly.

Line 438: displayed a wild-type pattern of ...

It was changed accordingly.

Line 505: formation of cadherin-catenin adhesion ...

It was changed accordingly.

Line 688: , and RNA processing factors were identified (Table S4).

It was changed accordingly.

Line 691: in the testes of males with belKD ...

It was changed accordingly.

Line 827: We suggest that ...

It was changed accordingly.

Line 831: Assuming that Belle is an upstream ...

It was changed accordingly.

Line 844: accomplished by these ligands is critical ...

It was changed accordingly.

Reviewer 2 Report

Kotov et al. use Drosophila to investigate the role of the DDX3 fly orthologue belle in testicular development and cancer. Their main findings are: 1. belle downregulation in somatic stem cells causes improper segregation of the cyst- from the germ cells, leading to niche disruption and consistent accumulation of tumour-like clusters; 2. adhesion restoration by beta-integrin expression (and not by cadherin/catenin complexes expression) in the belle-KD context rescues pre-meiotic defects and tumour-like phenotypes; 3. somatic expression of some Belle targets identified by CLIPseq is sufficient as to partially rescue the tumour-like phenotype in a belle-KD context.

Altogether, these data shed light on some central mechanisms governing the interplay between somatic and germ stem lineages during gamete production and differentiation in the fly male gonads. Given the functional conservation of DDX3 in humans, these findings may be used to clarify the aberrant signalling intervening between the germ and somatic Sertoli cells in testicular cancer.

The study is interesting and well designed, however some modifications and implementations are necessary before being accepted for publication.

A main concern is the use of the c587-Gal4 strain as a control. Each time one is inducing the KD of a given gene by RNAi, it is mandatory to use as a control the same GAL4 element driving an irrelevant hairpin, such as GFP, RFP or Luc RNAi, so as to exclude important side effects due to the boosting of the RNAi machinery itself. The authors are not asked to repeat all the experiments, they should just replicate panels C and D of Figure 1 using c587>Luc/GFP/RFP RNAi as a control and include images and statistics in the rebuttal. Please check the efficacy of all the RNAi lines used in the study indicating the percent decrease by citing other studies which used and validated them or by IF, WB or RT-PCR.

Figure 1:

change “wild type control” to “c587-Gal4” in C. despite the difference in cyst stem cell accumulation does not seem to be significant (F), the Tj-positive cells in D and E are much more in number (and much bigger in nuclear size) than those in C, and this is confusing. I appreciate the variability of a growing tumour often impedes obtaining statistically significant data, but authors should however mention in the text that belle KD in the cyst stem cells promotes the growth of tumour-like structures composed of a large fraction of germ cells (non cell-autonomous effect) and a minor fraction of cyst cells (cell-autonomous effect). as this study is first characterisation of belle KD in the somatic stem lineage of the testis, the authors should also try to motivate the cell-autonomous phenotype of belle depletion, i.e. nuclear size increase. Is there any explanation for this in the literature? Are those cells undergoing endocycle? Moreover, the authors show cyst cells’ morphology in Fig. S1, but they do not discuss it. Is it a mere consequence of cell-cell adhesion disruption? Please add some information about this in the main text.

Figure 2:

change “wild type control” to “c587-Gal4” in A and C. it is quite strange to find such a low level of PH3 positivity throughout the germ tumour masses; how are they expected to expand if not by dividing? I do not mean authors’ results are not reliable, but they should discuss this briefly in the text.

Figure S2:

While the driver indicated beside the panels is tj, the figure legend reports c587-Gal4, please fix this.

Figure S3:

Change “control” to “c587-Gal4” in graph A.

Figure 3:

Change “spectrocomes” to “spectrosomes” beside the graph in F.

Figure 5:

Please provide quantification of the bands shown in B, as results are not intuitive for Caf1-55 and Rho1.

Text:

I suggest to modify the title as follows:

“The Drosophila RNA helicase Belle (DDX3) non-autonomously suppresses germ line tumorigenesis via regulation of a specific mRNA set”

line 64: Dishevelled

line 83: “by” instead of “via

line 110: please change to: “enter meiosis”

Line 455: It is necessary a brief introduction, like this: “The ARGONAUTE family protein Piwi is known to regulate germ stem cell proliferation in Drosophila testes (48). Analysing its expression, we observed that…”

line 505: catenin instead of katenin

line 764: mediate instead of mediates

line 822: please add “(not shown)” after “caf1-55KD”

line 831: please add “an” before “upstream”

line 844: please remove “a” at the end of the line

Finally, I suggest to remove the last sentence of the abstract as, although plausible, is it merely speculative.

Author Response

Author's Reply to the Review Reports (Reviewer 2)

Reviewer 2

Comments and Suggestions for Authors

Kotov et al. use Drosophila to investigate the role of the DDX3 fly orthologue belle in testicular development and cancer. Their main findings are: 1. belle downregulation in somatic stem cells causes improper segregation of the cyst- from the germ cells, leading to niche disruption and consistent accumulation of tumour-like clusters; 2. adhesion restoration by beta-integrin expression (and not by cadherin/catenin complexes expression) in the belle-KD context rescues pre-meiotic defects and tumour-like phenotypes; 3. somatic expression of some Belle targets identified by CLIPseq is sufficient as to partially rescue the tumour-like phenotype in a belle-KD context.

Altogether, these data shed light on some central mechanisms governing the interplay between somatic and germ stem lineages during gamete production and differentiation in the fly male gonads. Given the functional conservation of DDX3 in humans, these findings may be used to clarify the aberrant signalling intervening between the germ and somatic Sertoli cells in testicular cancer.

The study is interesting and well designed, however some modifications and implementations are necessary before being accepted for publication.

 Thank you for your very helpful comments. We have incorporated our responses in the revised version of the paper, which has clearly improved the manuscript.

A main concern is the use of the c587-Gal4 strain as a control. Each time one is inducing the KD of a given gene by RNAi, it is mandatory to use as a control the same GAL4 element driving an irrelevant hairpin, such as GFP, RFP or Luc RNAi, so as to exclude important side effects due to the boosting of the RNAi machinery itself. The authors are not asked to repeat all the experiments, they should just replicate panels C and D of Figure 1 using c587>Luc/GFP/RFP RNAi as a control and include images and statistics in the rebuttal.

Yes, we used line y[1] v[1]; P{y[+t7.7] v[+t1.8]=TRiP.HMS00017}attP2 (#33623, Bloomington) for raising flies c587-GAL4>UAS-white RNAi as independent control and included images and statistics in the manuscript revision (see new Figure S1).

 Please check the efficacy of all the RNAi lines used in the study indicating the percent decrease by citing other studies which used and validated them or by IF, WB or RT-PCR.

It is possible to quantify a level of transgene expression in the case of using cells of the same type, for example, cell culture. However, Drosophila testes consist of multiple populations of different somatic and germ cells, with different cell numbers (Figure 1A). Note, that the number of early cyst cells per testis is significantly less than the number of germ cells. Since belle is normally expressed ubiquitously, it is very problematic to quantify the drop in belle expression in the case of knockdown only in the cyst cell population by Western blot, RT-qPCR or immunostaining (the last one is a non-quantitative method, and in our case belKD leads to a significant violation of whole testis morphology and to change in the shape of cyst cells from a flat elongated shape to a spherical one). We should note, however, that the UAS-bel RNAi line (VDRC #6299) has proven effectiveness and is in widespread use (Dietzl et al., 2007, Nature; Poulton et al., 2011; Development; Kotov et al., 2016, Mol Biol Cell; Mummery-Widmer et al., 2009, Nature; Lee et al., 2016, Cell; Cusmano et al., 2019, Front. Physiol, etc.). To obtain evidence for driver-activated expression of the transgenes in the rescue experiments, we performed Western blot analysis using anti-Arm and anti-beta-integrin antibodies and found that in both rescue experiments the levels of these proteins were increased. We included these data in (Fig. S4C and Fig. S5A). Note that expression of transgenic Arm copy did not lead to the restoration of early stages of spermatogenesis in the background of belKD in cyst cells, but expression of beta-integrin did. In addition, although we cannot quantify the level of bel knockdown, we have shown that male fertility was not restored by over-expression of beta-integrin, indicating other disorders caused by belKD in somatic cyst cells (Fig. S5B, C). Other RNAi lines (for Belle target genes) were received from the VDRC collection and each of them has been used in at least three published studies and has proven effectiveness.

Figure 1: change “wild type control” to “c587-Gal4” in C.

It was changed accordingly.

despite the difference in cyst stem cell accumulation does not seem to be significant (F), the Tj-positive cells in D and E are much more in number (and much bigger in nuclear size) than those in C, and this is confusing. I appreciate the variability of a growing tumour often impedes obtaining statistically significant data, but authors should however mention in the text that belle KD in the cyst stem cells promotes the growth of tumour-like structures composed of a large fraction of germ cells (non cell-autonomous effect) and a minor fraction of cyst cells (cell-autonomous effect).

Actually, the average number of Tj-positive cyst cells in case of belKD was found to be higher; however, in such testes we observed significant variations of this number among experimental testes compared to control ones (Fig. 1F). We changed the text to acknowledge the fact that belKD in cyst cells can lead to a cell-autonomous effect of accumulation of early cyst cells. We incorporated the sentence “Nevertheless, belKD in early cyst cells often leads to a cell-autonomous effect of their accumulation” in the final paragraph of 3.1 section of the Results.

 as this study is first characterisation of belle KD in the somatic stem lineage of the testis, the authors should also try to motivate the cell-autonomous phenotype of belle depletion, i.e. nuclear size increase. Is there any explanation for this in the literature? Are those cells undergoing endocycle? Moreover, the authors show cyst cells’ morphology in Fig. S1, but they do not discuss it. Is it a mere consequence of cell-cell adhesion disruption? Please add some information about this in the main text.

Since cyst cells do not surround germ cells in the belKD testes, they take a spherical shape, in contrast to the flat elongated shape in the cysts of the wild-type testes. Nuclei of cyst cells in the belKD testes also conform to a spherical shape. It is because they are not affected by the compressive forces that form the structure of the cyst in the wild type testes. We also observed a pronounced increase of nuclear volumes of cyst cells in the belKD testes and found a change in the morphology of the cyst cell nuclei: inside the nuclei Tj protein was excluded from a certain space, presumably from the nucleolus (Figure 1D,E). Both of these phenomena have not been described previously in the published literature. Apparently, in the absence of Belle expression in cyst cells, a cessation of their internal differentiation program takes place. However, we do not think that they become polyploid. We add some observations about this issue in the main text.

Figure 2: change “wild type control” to “c587-Gal4” in A and C.

It was changed accordingly.

 it is quite strange to find such a low level of PH3 positivity throughout the germ tumour masses; how are they expected to expand if not by dividing? I do not mean authors’ results are not reliable, but they should discuss this briefly in the text.

Because the meiotic cell division appears to be a rapid process compared to interphase, at any given moment we can see only a small number of dividing cells. This is in accordance with our previous unpublished observations. We included this note in the main text.

Figure S2: While the driver indicated beside the panels is tj, the figure legend reports c587-Gal4, please fix this.

This has been fixed. Here the testes of tj-GAL4>UAS-bel RNAi males were used for immunostaining (Figure S3A), whereas the testes of c587-GAL4>UAS-bel RNAi males were used for Western blot analysis (Figure S3C) .

Figure S3: Change “control” to “c587-Gal4” in graph A.

It was changed accordingly.

Figure 3: Change “spectrocomes” to “spectrosomes” beside the graph in F.

It was changed accordingly.

Figure 5: Please provide quantification of the bands shown in B, as results are not intuitive for Caf1-55 and Rho1.

It was done.

Text:

I suggest to modify the title as follows:

“The Drosophila RNA helicase Belle (DDX3) non-autonomously suppresses germ line tumorigenesis via regulation of a specific mRNA set”

It was changed accordingly.

line 64: Dishevelled

It was changed accordingly.

line 83: “by” instead of “via”

It was changed accordingly.

line 110: please change to: “enter meiosis”

It was changed accordingly.

Line 455: It is necessary a brief introduction, like this: “The ARGONAUTE family protein Piwi is known to regulate germ stem cell proliferation in Drosophila testes (48). Analysing its expression, we observed that…”

It was done.

line 505: catenin instead of katenin

It was changed accordingly.

line 764: mediate instead of mediates

It was changed accordingly.

line 822: please add “(not shown)” after “caf1-55KD”

We incorporated data for caf1-55KDs (new Figure S8, new Figure S9) and corresponding explanations in the text of the revised manuscript.

line 831: please add “an” before “upstream”

It was changed accordingly.

line 844: please remove “a” at the end of the line

It was changed accordingly.

Finally, I suggest to remove the last sentence of the abstract as, although plausible, is it merely speculative.

We wrote that “Failures in translation of a number of mRNA targets appear to additively contribute to developmental defects observed in the testes with belle knockdowns both in somatic cyst cells and in the germline”. We would like to leave this statement because we have included some additional data in the article to confirm it (see new Figure 6 and new Figure S8 and S9). We also provide the corresponding rationales in the Discussion. We slightly modified this statement as “By our hypothesis failures in the translation of a number of mRNA targets additively contribute to developmental defects observed in the testes with belle knockdowns both in cyst cells and in the germline”.

Round 2

Reviewer 2 Report

The authors have addressed my main concerns in a satisfactory way. In my opinion, the manuscript is now suitable for publication.